# OpenMask3D:
# Open-Vocabulary 3D Instance Segmentation

**Ayça Takmaz**[1*]          **Elisabetta Fedele**[1*]          **Robert W. Sumner**[1]

**Marc Pollefeys**[1,2]          **Federico Tombari**[3]          **Francis Engelmann**[1,3]

[1]ETH Zürich          [2]Microsoft          [3]Google

openmask3d.github.io

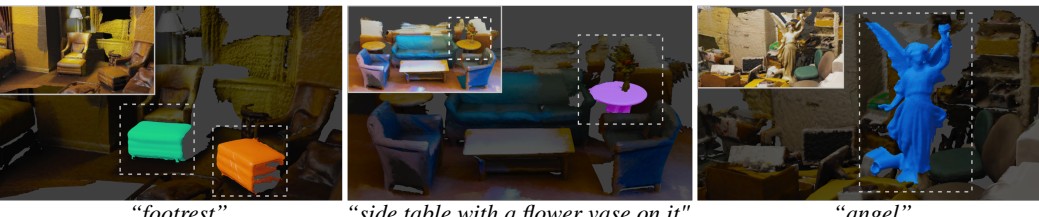

*"footrest"*          *"side table with a flower vase on it"*          *"angel"*

**Figure 1: Open-Vocabulary 3D Instance Segmentation.** Given a 3D scene *(top)* and free-form user queries *(bottom)*, our OpenMask3D segments object instances and scene parts described by the open-vocabulary queries.

## Abstract

We introduce the task of open-vocabulary 3D *instance* segmentation. Current approaches for 3D instance segmentation can typically only recognize object categories from a pre-defined closed set of classes that are annotated in the training datasets. This results in important limitations for real-world applications where one might need to perform tasks guided by novel, open-vocabulary queries related to a wide variety of objects. Recently, open-vocabulary 3D scene understanding methods have emerged to address this problem by learning queryable features for each point in the scene. While such a representation can be directly employed to perform *semantic* segmentation, existing methods cannot separate multiple object *instances*. In this work, we address this limitation, and propose OpenMask3D, which is a zero-shot approach for open-vocabulary 3D *instance* segmentation. Guided by predicted class-agnostic 3D instance masks, our model aggregates per-mask features via multi-view fusion of CLIP-based image embeddings. Experiments and ablation studies on ScanNet200 and Replica show that OpenMask3D outperforms other open-vocabulary methods, especially on the long-tail distribution. Qualitative experiments further showcase OpenMask3D's ability to segment object properties based on free-form queries describing geometry, affordances, and materials.

## 1   Introduction

3D instance segmentation, which is the task of predicting 3D object instance masks along with their object categories, has many crucial applications in fields such as robotics and augmented reality. Due to its significance, 3D instance segmentation has been receiving a growing amount of attention in recent years. Despite the remarkable progress made in 3D instance segmentation methods [8, 9, 39, 58, 77, 78], it is noteworthy that these methods operate under a closed-set paradigm, where the set of object categories is limited and closely tied to the datasets used during training.

We argue that there are two key problems with *closed-vocabulary* 3D instance segmentation. First, these approaches are limited in their ability to understand a scene beyond the object categories seen during training. Despite the significant success of 3D instance segmentation approaches from recent years, these closed-vocabulary approaches may fail to recognize and correctly classify novel objects. One of the main advantages of open-vocabulary approaches, is their ability to zero-shot learn

---

[*] Equal contribution.

37th Conference on Neural Information Processing Systems (NeurIPS 2023).

categories that are not present at all in the training set. This ability has potential benefits for many applications in fields such as robotics, augmented reality, scene understanding and 3D visual search. For example, it is essential for an autonomous robot to be able to navigate in an unknown environment where novel objects can be present. Furthermore, the robot may need to perform an action based on a free-form query, such as "find the side table with a flower vase on it", which is challenging to perform with the existing closed-vocabulary 3D instance segmentation methods. Hence, the second key problem with closed-vocabulary approaches is their inherent limitation to recognize only object classes that are predefined at training time.

In an attempt to address and overcome the limitations of a closed-vocabulary setting, there has been a growing interest in open-vocabulary approaches. A line of work [20, 43, 45] investigates open-vocabulary 2D image segmentation task. These approaches are driven by advances in large-scale model training, and largely rely on recent foundation models such as CLIP [55] and ALIGN [33] to obtain text-image embeddings. Motivated by the success of these 2D open-vocabulary approaches, another line of work has started exploring 3D open-vocabulary scene understanding task [24, 32, 52], based on the idea of lifting image features from models such as CLIP [55], LSeg [43], and OpenSeg [20] to 3D. These approaches aim to obtain a task-agnostic feature representation for each 3D point in the scene, which can be used to query concepts with open-vocabulary descriptions such as object semantics, affordances or material properties. Their output is typically a heatmap over the points in the scene, which has limited applications in certain aspects, such as handling object instances.

In this work, we propose OpenMask3D, an open-vocabulary 3D instance segmentation method which has the ability to reason beyond a predefined set of concepts. Given an RGB-D sequence, and the corresponding 3D reconstructed geometry, OpenMask3D predicts 3D object instance masks, and computes a *mask-feature* representation. Our two-stage pipeline consists of a class-agnostic mask proposal head, and a mask-feature aggregation module. Guided by the predicted class-agnostic 3D instance masks, our mask-feature aggregation module first finds the frames in which the instances are highly visible. Then, it extracts CLIP features from the best images of each object mask, in a multi-scale and crop-based manner. These features are then aggregated across multiple views to obtain a feature representation associated with each 3D instance mask. Our approach is intrinsically different from the existing 3D open-vocabulary scene understanding approaches [24, 32, 52] as we propose an instance-based feature computation approach instead of a point-based one. Computing a *mask-feature* per object instance enables us to retrieve object instance masks based on their similarity to any given query, equipping our approach with open-vocabulary 3D instance segmentation capabilities. As feature computation is performed in a zero-shot manner, OpenMask3D is capable of preserving information about novel objects as well as long-tail objects better, compared to trained or fine-tuned counterparts. Furthermore, OpenMask3D goes beyond the limitations of a closed-vocabulary paradigm, and allows segmentation of object instances based on free-form queries describing object properties such as semantics, geometry, affordances, and material properties.

Our contributions are three-fold:

- We introduce the open-vocabulary 3D instance segmentation task in which the object instances that are similar to a given text-query are identified.

- We propose OpenMask3D, which is the first approach that performs open-vocabulary 3D instance segmentation in a zero-shot manner.

- We conduct experiments to provide insights about design choices that are important for developing an open-vocabulary 3D instance segmentation model.

## 2 Related work

**Closed-vocabulary 3D semantic and instance segmentation.** Given a 3D scene as input, 3D semantic segmentation task aims to assign a semantic category to each point in the scene [2–4, 9, 14, 15, 22, 28, 29, 31, 38, 42, 44, 46, 47, 53, 54, 63, 64, 66, 68, 70, 73]. 3D instance segmentation goes further by distinguishing multiple objects belonging to the same semantic category, predicting individual masks for each object instance [13, 16, 25, 27, 34, 41, 58, 62, 65, 67, 74]. The current state-of-the-art approach on the ScanNet200 benchmark [10, 57] is Mask3D [58], which leverages a transformer architecture to generate 3D mask proposals together with their semantic labels. However, similar to existing methods, it assumes a limited set of semantic categories that can be assigned to an object instance. In particular, the number of labels is dictated by the annotations provided in the training datasets, 200 – in the case of ScanNet200 [57]. Given that the English language encompasses

numerous nouns, in the order of several hundred thousand [69], it is clear that existing closed-vocabulary approaches have an important limitation in handling object categories and descriptions.

**Foundation models.** Recent multimodal foundation models [1, 7, 21, 33, 55, 75, 76] leverage large-scale pretraining to learn image representations guided by natural language descriptions. These models enable zero-shot transfer to various downstream tasks such as object recognition and classification. Driven by the progress in large-scale model pre-training, similar foundation models for images were also explored in another line of work [5, 51], which aim to extract class-agnostic features from images. Recently, steps towards a foundation model for image segmentation were taken with SAM [36]. SAM has the ability to generate a class-agnostic 2D mask for an object instance, given a set of points that belong to that instance. This capability is valuable for our approach, especially for recovering high-quality 2D masks from projected 3D instance masks, as further explained in Sec. 3.2.2.

**Open vocabulary 2D segmentation.** As large vision-language models gained popularity for image classification, numerous new approaches [11, 20, 23, 26, 32, 40, 43, 45, 48, 56, 71, 72, 79, 80] have emerged to tackle open-vocabulary or zero-shot image semantic segmentation. One notable shift was the transition from image-level embeddings to pixel-level embeddings, equipping models with localization capabilities alongside classification. However, methods with pixel-level embeddings, such as OpenSeg [20] and OV-Seg [45], strongly rely on the accuracy of 2D segmentation masks, and require a certain degree of training. In our work, we rely on CLIP [55] features without performing finetuning or any additional training, and compute 2D masks using the predicted 3D instance masks.

**Open-vocabulary 3D scene understanding.** Recent success of 2D open-vocabulary segmentation models such as LSeg [43], OpenSeg [20], and OV-Seg[45] has motivated researchers in the field of 3D scene understanding to explore the open vocabulary setting [6, 12, 19, 24, 30, 32, 35, 37, 49, 52, 59, 60]. OpenScene [52] uses per-pixel image features extracted from posed images of a scene and obtains a point-wise task-agnostic scene representation. On the other hand, approaches such as LERF [35] and DFF [37] leverage the interpolation capabilities of NeRFs [50] to extract a semantic field of the scene. However, it is important to note that all of these approaches have a limited understanding of object *instances* and inherently face challenges when dealing with instance-related tasks.

# 3 Method

**Overview.** Our OpenMask3D model is illustrated in Fig. 2. Given a set of posed RGB-D images captured in a scene, along with the reconstructed scene point cloud ①, OpenMask3D predicts 3D instance masks with their associated per-mask feature representations, which can be used for querying instances based on open-vocabulary concepts ④. Our OpenMask3D has two main building blocks: a *class agnostic mask proposal head* ② and a *mask-feature computation module* ③. The class-agnostic mask proposal head predicts binary instance masks over the points in the point cloud. The mask-feature computation module leverages pre-trained CLIP [55] vision-language model in order to compute meaningful and flexible features for each mask. For each proposed instance mask, the mask-feature computation module first selects the views in which the 3D object instance is highly visible. Subsequently, in each selected view, the module computes a 2D segmentation mask guided by the projection of the 3D instance mask, and refines the 2D mask using the SAM [36] model. Next, the CLIP encoder is employed to obtain image-embeddings of multi-scale image-crops bounding the computed 2D masks. These image-level embeddings are then aggregated across the selected frames in order to obtain a mask-feature representation. Sec. 3.1 describes the class agnostic mask proposal head, and Sec. 3.2 describes the mask-feature computation module.

The key novelty of our method is that it follows an *instance-mask oriented* approach, contrary to existing 3D open-vocabulary scene understanding models which typically compute *per-point* features. These point-feature oriented models have inherent limitations, particularly for identifying object *instances*. Our model aims to overcome such limitations by introducing a framework that employs class agnostic instance masks and aggregates informative features for each object *instance*.

**Input.** Our pipeline takes as input a collection of posed RGB-D images captured in an indoor scene, and the reconstructed point cloud representation of the scene. We assume known camera parameters.

## 3.1 Class agnostic mask proposals

The first step of our approach involves generating $M$ class-agnostic 3D mask proposals $\mathbf{m}_1^{\mathbf{3D}}, \ldots, \mathbf{m}_M^{\mathbf{3D}}$. Let $\mathbf{P} \in \mathbb{R}^{P \times 3}$ denote the point cloud of the scene, where each 3D point is represented with its corresponding 3D coordinates. Each 3D mask proposal is represented by a

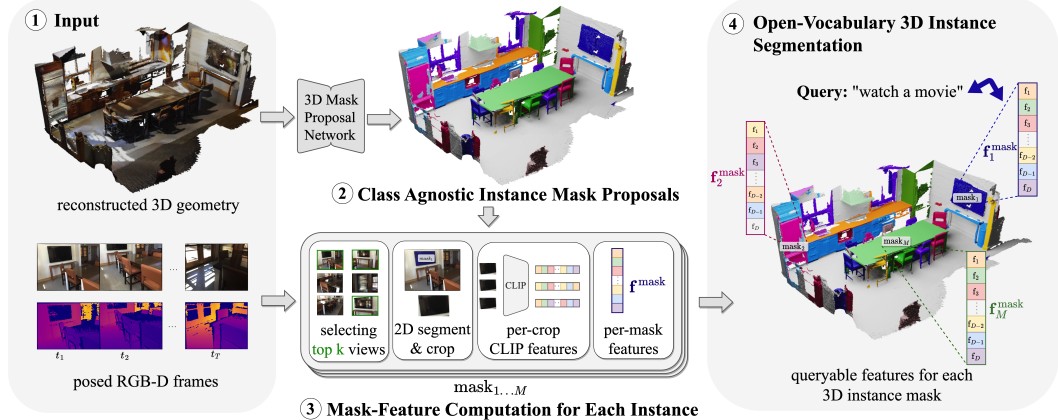

**Figure 2: An overview of our approach.** We propose OpenMask3D, the first open-vocabulary 3D instance segmentation model. Our pipeline consists of four subsequent steps: ① Our approach takes as input posed RGB-D images of a 3D indoor scene along with its reconstructed point cloud. ② Using the point cloud, we compute class-agnostic instance mask proposals. ③ Then, for each mask, we compute a feature representation. ④ Finally, we obtain an open-vocabulary 3D instance segmentation representation, which can be used to retrieve objects related to queried concepts embedded in the CLIP [55] space.

binary mask $\mathbf{m_i^{3D}} = (m_{i1}^{3D}, \dots, m_{iP}^{3D})$ where $m_{ij}^{3D} \in \{0, 1\}$ indicates whether the $j$-th point belongs to $i$-th object instance. To generate these masks, we leverage the transformer-based mask-module of a pre-trained 3D instance segmentation model [58], which is frozen during our computations. The architecture consists of a sparse convolutional backbone based on the MinkowskiUNet [9], and a transformer decoder. Point features obtained from the feature backbone are passed through the transformer decoder, which iteratively refines the instance queries, and predicts an instance heatmap for each query. In the original setup, [58] produces two outputs: a set of $M$ binary instance masks obtained from the predicted heatmaps, along with predicted class labels (from a predefined closed set) for each mask. In our approach, we adapt the model to exclusively utilize the binary instance masks, discarding the predicted class labels and confidence scores entirely. These *class-agnostic* binary instance masks are then utilized in our mask-feature computation module, in order to go beyond semantic class predictions limited to a closed-vocabulary, and obtain open-vocabulary representations instead. Further details about the class-agnostic mask proposal module are provided in Appendix A.1.

### 3.2   Mask-feature computation module

Mask-feature computation module aims to compute a task-agnostic feature representation for each predicted instance mask obtained from the class-agnostic mask proposal module. The purpose of this module is to compute a feature representation that can be used to query open-vocabulary concepts. As we intend to utilize the CLIP text-image embedding space and maximally retain information about long-tail or novel concepts, we solely rely on the CLIP visual encoder to extract image-features on which we build our mask-features.

As illustrated in Fig. 3, the mask-feature computation module consists of several steps. For each instance mask-proposal, we first compute the visibility of the object instance in each frame of the RGB-D sequence, and select the top-$k$ views with maximal visibility. In the next step, we compute a 2D object mask in each selected frame, which is then used to obtain multi-scale image-crops in order to extract effective CLIP features. The image-crops are then passed through the CLIP visual encoder to obtain feature vectors that are average-pooled over each crop and each selected view, resulting in the final mask-feature representation. In Sec. 3.2.1, we describe how we select a set of top-$k$ frames for each instance. In Sec. 3.2.2, we describe how we crop the frames, based on the object instance we want to embed. In Sec. 3.2.3, we describe how we compute the final mask-features per object.

### 3.2.1   Frame selection

Obtaining representative *images* of the proposed object instances is crucial for extracting accurate CLIP features. To achieve this, we devise a strategy to select, for each of the $M$ predicted instances, a subset of representative frames (Fig. 3, ⓐ) from which we extract CLIP features. In particular, our devised strategy selects frames based on their visibility scores $s_{ij}$ for each mask $i$ in each view $j$. Here, we explain how we compute these visibility scores.

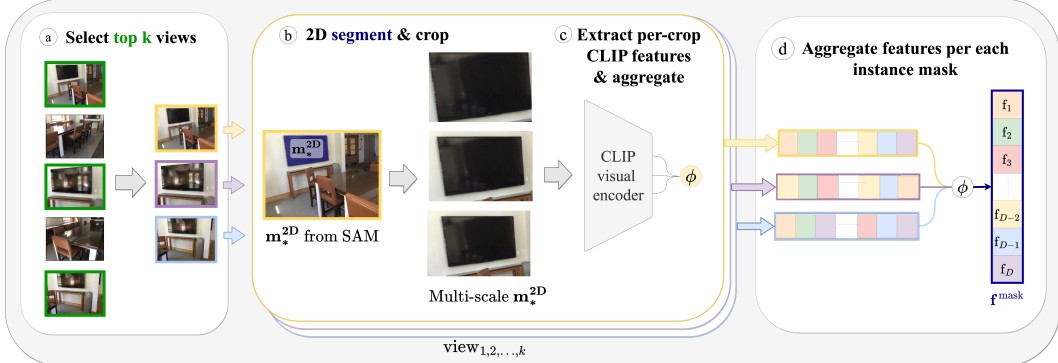

**Figure 3: Mask-Feature Computation Module.** For each instance mask, ⓐ we first compute the visibility of the instance in each frame, and select top-$k$ views with maximal visibility. In ⓑ, we compute a 2D object mask in each selected frame, which is used to obtain multi-scale image-crops in order to extract effective CLIP features. ⓒ The image-crops are then passed through the CLIP visual encoder to obtain feature vectors that are average-pooled over each crop and ⓓ each selected view, resulting in the final mask-feature representation.

Given a mask proposal $\mathbf{m_i^{3D}}$ corresponding to the $i$-th instance, we compute the visibility score $s_{ij} \in [0, 1]$ for the $j$-th frame using the following formula:

$$s_{ij} = \frac{vis(i,j)}{\max_{j'} \left( vis(i,j') \right)}$$

Here, $vis(i,j)$ represents the number of points from mask $i$ that are visible in frame $j$. Note that we assume that each mask is visible in at least one frame. We compute $vis(i,j)$ using the following approach. First, for each 3D point from the $i$-th instance mask, we determine whether the point appears in the camera's field of view (FOV) in the $j$-th frame. To achieve this, we use intrinsic ($\mathbf{I}$) and extrinsic ($\mathbf{R}|\mathbf{t}$) matrices of the camera from that frame to project each 3D point in the point cloud to the image plane. We obtain corresponding 2D homogeneous coordinates $\mathbf{p_{2D}} = (u, v, w)^\top = \mathbf{I} \cdot (\mathbf{R}|\mathbf{t}) \cdot \mathbf{x}$, where $\mathbf{x} = (x, y, z, 1)^\top$ is a point represented in homogeneous coordinates. We consider a point to be in the camera's FOV in the $j$-th frame if $w \neq 0$, and the $\frac{u}{w}$ value falls within the interval $[0, W-1]$ while $\frac{v}{w}$ falls within $[0, H-1]$. Here $W$ and $H$ represent the width and height of the image, respectively. Next, it is important to note that 3D points that are in the camera's FOV are not necessarily visible from a given camera pose, as they might be occluded by other parts of the scene. Hence, we need to check whether the points are in fact *visible*. In order to examine whether a point is occluded from a given camera viewpoint, we compare the depth value of the 3D point deriving from the geometric computation (i.e. $w$) and the measured depth, i.e. depth value of the projected point in the depth image. We consider points that satisfy the inequality $w - d > k_{\text{threshold}}$ as occluded, where $k_{\text{threshold}}$ is a hyper-parameter of our method. If the projection of a point from mask $i$ is not occluded in the $j$-th view, it contributes to the visible point count, $vis(i,j)$. Once we compute mask-visibility scores for each frame, we select the top $k_{\text{view}}$ views with the highest scores $s_{ij}$ for each mask $i$. Here, $k_{\text{view}}$ represents another hyperparameter.

### 3.2.2 2D mask computation and multi-scale crops

In this section, we explain our approach for computing CLIP features based on the frames selected in the previous step. Given a selected frame for a mask, our goal is to find the optimal image crops from which to extract features, as shown in (Fig. 3, ⓑ). Simply considering all of the projected points of the mask often results in imprecise and noisy bounding boxes, largely affected by the outliers (please see Appendix A.2.2). To address this, we use a class-agnostic 2D segmentation model, SAM [36], which predicts a 2D mask conditioned on a set of input points, along with a mask confidence score.

SAM is sensitive to the set of input points (see Appendix A.2.3, A.2.4). Therefore, to obtain a high quality mask from SAM with a high confidence score, we take inspiration from the RANSAC algorithm [18] and proceed as described in Algorithm 1. We sample $k_{sample}$ points from the projected points, and run SAM using these points. The output of SAM at a given iteration $r$ is a 2D mask ($\mathbf{m_r^{2D}}$) and a mask confidence score ($\texttt{score}_r$). This process is repeated for $k_{rounds}$ iterations, and the 2D mask with the highest confidence score is selected.

Next, we use the resulting mask $\mathbf{m_*^{2D}}$ to generate $L = 3$ multi-level crops of the selected image. This allows us to enrich the features by incorporating more contextual information from the surroundings.

**Algorithm 1** - 2D mask selection algorithm

---

$\texttt{score}_* \leftarrow 0, \mathbf{m}_*^{\mathbf{2D}} \leftarrow \mathbf{0}, r \leftarrow 0$
**while** $r < k_{rounds}$ **do**
    Sample $k_{sample}$ points among the projected points at random
    Compute the mask $\mathbf{m}_{\mathbf{r}}^{\mathbf{2D}}$ and the score $\texttt{score}_r$ based on the sampled points using SAM
    **if** $\texttt{score}_r > \texttt{score}_*$ **then**
        $\texttt{score}_* \leftarrow \texttt{score}_r, \mathbf{m}_*^{\mathbf{2D}} \leftarrow \mathbf{m}_{\mathbf{r}}^{\mathbf{2D}}$
    **end if**
    $r \leftarrow r + 1$
**end while**

---

Specifically, the first bounding box $\mathbf{b^1} = (x_1^1, y_1^1, x_2^1, y_2^1)$ with $0 \leq x_1^1 < x_2^1 < W$ and $0 \leq y_1^1 < y_2^1 < H$ is a tight bounding box derived from the 2D mask. The other bounding boxes $\mathbf{b^2}$ and $\mathbf{b^3}$ are incrementally larger. In particular, the 2D coordinates of $\mathbf{b^i}$ for $i = 2, 3$ are obtained as follows.

$$x_1^l = \max(0, x_1^1 - (x_2^1 - x_1^1) \cdot k_{exp} \cdot l)$$
$$y_1^l = \max(0, y_1^1 - (y_2^1 - y_1^1) \cdot k_{exp} \cdot l)$$
$$x_2^l = \min(x_2^1 + (x_2^1 - x_1^1) \cdot k_{exp} \cdot l, W - 1)$$
$$y_2^l = \min(y_2^1 + (y_2^1 - y_1^1) \cdot k_{exp} \cdot l, H - 1)$$

Note that here $l$ represents the level of the features and $k_{exp} = 0.1$ is a predefined constant.

### 3.2.3 CLIP feature extraction and mask-feature aggregation

For each instance mask, we collect $k \cdot L$ images by selecting top-$k$ views and obtaining $L$ multi-level crops as described in Sec. 3.2.1 and Sec. 3.2.2. Collected image crops are then passed through the CLIP visual encoder in order to extract image features in the CLIP embedding space, as shown in (Fig. 3, ©). We then aggregate the features obtained from each crop that correspond to a given instance mask in order to get an average per-mask CLIP feature (Fig. 3, ⓓ). The computed features are task-agnostic, and can be used for various instance-based tasks by encoding a given text or image-based query, using the same CLIP model we employed to encode the image crops.

## 4 Experiments

In this section, we present quantitative and qualitative results from our method OpenMask3D. In Sec 4.1, we quantitatively evaluate our method, and compare OpenMask3D with supervised 3D instance segmentation approaches as well as existing open-vocabulary 3D scene understanding models we adapted for the 3D instance segmentation task. Furthermore, we provide an ablation study for OpenMask3D. In Sec. 4.2, we share qualitative results for open-vocabulary 3D instance segmentation, demonstrating potential applications. Additional results are provided in the Appendix.

### 4.1 Quantitative results: closed-vocabulary 3D instance segmentation evaluation

We evaluate our approach on the closed-vocabulary 3D instance segmentation task. We conduct additional experiments to assess the generalization capability of OpenMask3D.

#### 4.1.1 Experimental setting

**Data.** We conduct our experiments using the ScanNet200 [57] and Replica [61] datasets. We report our ScanNet200 results on the validation set consisting of 312 scenes, and evaluate for the 3D instance segmentation task using the closed vocabulary of 200 categories from the ScanNet200 annotations. Rozenberszki et al. [57] also provide a grouping of ScanNet200 categories based on the frequency of the number of labeled surface points in the training set, resulting in 3 subsets: *head* (66 categories), *common* (68 categories), *tail* (66 categories). This grouping enables us to evaluate the performance of our method on the long-tail distribution, making ScanNet200 a natural choice as an evaluation dataset. To assess the generalization capability of our method, we further experiment with the Replica [61] dataset, and evaluate on the *office0, office1, office2, office3, office4, room0, room1, room2* scenes.

**Metrics.** We employ a commonly used 3D instance segmentation metric, average precision (AP). AP scores are evaluated at mask overlap thresholds of $50\%$ and $25\%$, and averaged over the overlap

| Model | Image Features | AP | AP$_{50}$ | AP$_{25}$ | head (AP) | common (AP) | tail (AP) |
|---|---|---|---|---|---|---|---|
| *Closed-vocabulary, fully supervised* | | | | | | | |
| Mask3D [58] | - | 26.9 | 36.2 | 41.4 | 39.8 | 21.7 | 17.9 |
| *Open-vocabulary* | | | | | | | |
| OpenScene [52] (2D Fusion) + masks | OpenSeg [20] | 11.7 | 15.2 | 17.8 | 13.4 | 11.6 | 9.9 |
| OpenScene [52] (3D Distill) + masks | OpenSeg [20] | 4.8 | 6.2 | 7.2 | 10.6 | 2.6 | 0.7 |
| OpenScene [52] (2D/3D Ens.) + masks | OpenSeg [20] | 5.3 | 6.7 | 8.1 | 11.0 | 3.2 | 1.1 |
| OpenScene [52] (2D Fusion) + masks | LSeg [43] | 6.0 | 7.7 | 8.5 | 14.5 | 2.5 | 1.1 |
| OpenMask3D (Ours) | CLIP [55] | **15.4** | **19.9** | **23.1** | **17.1** | **14.1** | **14.9** |

**Table 1: 3D instance segmentation results on the ScanNet200 validation set.** Metrics are respectively: AP averaged over an overlap range, and AP evaluated at 50% and 25% overlaps. We also report AP scores for head, common, tail subsets of ScanNet200. Mask3D [58] is fully-supervised, while OpenScene [52] is built upon on 2D models (LSeg [43], OpenSeg [20]) trained on labeled datasets for 2D semantic segmentation. Since OpenScene [52] does not provide instance masks, we aggregate its per-point features using class-agnostic masks. OpenMask3D outperforms other open-vocabulary counterparts, particularly on the long-tail classes.

range of $[0.5 : 0.95 : 0.05]$ following the evaluation scheme from ScanNet [10]. Computation of the metrics requires each mask to be assigned a prediction confidence score. We assign a prediction confidence score of $1.0$ for each predicted mask in our experiments.

**OpenMask3D implementation details**. We use posed RGB-depth pairs for both the ScanNet200 and Replica datasets, and we process 1 frame in every 10 frames in the RGB-D sequences. In order to compute image features on the mask-crops, we use CLIP [55] visual encoder from the ViT-L/14 model pre-trained at a 336 pixel resolution, which has a feature dimensionality of 768. For the visibility score computation, we use $k_{threshold} = 0.2$, and for top-view selection we use $k_{view} = 5$. In all experiments with multi-scale crops, we use $L = 3$ levels. In the 2D mask selection algorithm based on SAM [36], we repeat the process for $k_{rounds} = 10$ rounds, and sample $k_{sample} = 5$ points at each iteration. For the class-agnostic mask proposals, we use the Mask3D [58] model trained on ScanNet200 instances, and exclusively use the predicted binary instance masks in our pipeline. We do not filter any instance mask proposals, and run DBSCAN [17] to obtain spatially contiguous clusters, breaking down masks into smaller new masks when necessary. This process results in a varying number of mask proposals for each scene. For further implementation details about OpenMask3D, please refer to Appendix A. Computation of the mask-features of a ScanNet scene on a single GPU takes 5-10 minutes depending on the number of mask proposals and number of frames in the RGB-D sequence. Note that once the per-mask features of the scene are computed, objects can be queried in real-time (~ 1-2 ms) with arbitrary open-vocabulary queries.

**Methods in comparison.** We compare with Mask3D [58], which is the current state-of-the-art on the ScanNet200 3D instance segmentation benchmark. We also compare against recent open-vocabulary 3D scene understanding model variants (2D Fusion, 3D Distill, 2D/3D Ensemble) from OpenScene [52]. Note that since OpenScene only generates a per-point feature vector (without any per-instance aggregation), it is not possible to directly compare it with our method. To address this, we extended OpenScene by averaging its per-point features within each instance mask generated by our mask module (see Appendix C for more details).

**Class assignment.** Our open-vocabulary approach does *not* directly predict semantic category labels per each instance mask, but it instead computes a task-agnostic feature vector for each instance, which can be used for performing a semantic label assignment. In order to evaluate our model on the closed-vocabulary 3D instance segmentation task, we need to assign each object instance to a semantic category. Similar to OpenScene [52], we compute cosine similarity between mask-features and the text embedding of a given query in order to perform class assignments. Following Peng et al. [52], we use prompts in the form of "a {} in a scene", and compute text-embeddings using CLIP model ViT-L/14(336px) [55] for each semantic class in the ScanNet200 dataset. This way, we compute a similarity score between each instance and each object category, and assign instances to the category with the closest text embedding.

### 4.1.2 Results and analysis

**3D closed-vocabulary instance segmentation results.** We quantitatively evaluate our approach on the closed-vocabulary instance segmentation task on the ScanNet200 [10, 57] and Replica [61] datasets. Closed-vocabulary instance segmentation results are provided in Tab. 1 and Tab. 2.

State-of-the-art fully supervised approach Mask3D [58] demonstrates superior performance compared to open-vocabulary counterparts. While this gap is more prominent for the *head* and *common* categories, the difference is less apparent for the *tail* categories. This outcome is expected, as Mask3D benefits from full-supervision using the closed-set of class labels from the ScanNet200 dataset. Furthermore, as there are more training samples from the categories within the *head* and *common* subsets (please refer to [57] for the statistics), the fully-supervised approach is more frequently exposed to these categories - resulting in a stronger performance. When we compare OpenMask3D with other open-vocabulary approaches, we observe that our method performs better on 6 out of 6 metrics. Notably, OpenMask3D outperforms other open-vocabulary approaches especially on the *tail* categories by a significant margin. Our instance-centric method OpenMask3D, which is specifically designed for the open-vocabulary 3D instance segmentation task, shows stronger performance compared to other open-vocabulary approaches which adopt a point-centric feature representation. Differences between the feature representations are also illustrated in Fig. 5.

**How well does our method generalize?** As the mask module is trained on a closed-set segmentation dataset, ScanNet200 [57], we aim to investigate how well our method generalizes beyond the categories seen during training. Furthermore, we aim to analyze how well our method generalizes to out-of-distribution (OOD) data, such as scenes from another dataset, Replica [61]. To demonstrate the generalization capability of our approach, we conducted a series of experiments. First, we analyze the performance of our model when we use class-agnostic masks from a mask-predictor trained on the 20 original ScanNet classes [10], and evaluate on the ScanNet200 dataset. To evaluate how well our model performs on "unseen" categories, we group the ScanNet200 labels into two subsets: *base* and *novel* classes. We identify ScanNet200 categories that are semantically similar to the original ScanNet20 classes (e.g. chair and folded-chair, table and dining-table), resulting in 53 classes. We group all remaining object classes that are not similar to any class in ScanNet20 as "novel". In Tab. 3, we report results on seen ("base") classes, unseen ("novel") classes, and all classes. Our experiments show that the OpenMask3D variant using a mask proposal backbone trained on a smaller set of object annotations from ScanNet20 can generalize to predict object masks from a significantly larger set of objects (ScanNet200), resulting in only a marginal decrease in the performance. In a second experiment, we evaluate the performance of OpenMask3D on out-of-distribution data from Replica, using a mask predictor trained on ScanNet200. The results from this experiment, as presented in Tab. 2, indicate that OpenMask3D can indeed generalize to unseen categories as well as OOD data. OpenMask3D using a mask predictor module trained on a smaller set of objects seems to perform reasonably well in generalizing to various settings.

| Model | AP | $AP_{50}$ | $AP_{25}$ |
|---|---|---|---|
| OpenScene [52] (2D Fusion) + masks | 10.9 | 15.6 | 17.3 |
| OpenScene [52] (3D Distill) + masks | 8.2 | 10.5 | 12.6 |
| OpenScene [52] (2D/3D Ens.) + masks | 8.2 | 10.4 | 13.3 |
| OpenMask3D (Ours) | **13.1** | **18.4** | **24.2** |

**Table 2: 3D instance segmentation results on the Replica [61] dataset**. To assess how well our model generalizes to other datasets, we use instance masks from the mask proposal module trained on ScanNet200, and test it on Replica scenes. OpenMask3D outperforms other open-vocabulary counterparts on the Replica dataset.

| Method | Mask Training | Novel Classes | | | Base Classes | | | All Classes | |
|---|---|---|---|---|---|---|---|---|---|
| | | AP | $AP_{50}$ | $AP_{25}$ | AP | $AP_{50}$ | $AP_{25}$ | AP | tail (AP) |
| OpenScene [52] (2D Fusion) + masks | ScanNet20 | 7.6 | 10.3 | 12.3 | 11.1 | 15.0 | 17.7 | 8.5 | 6.1 |
| OpenScene [52] (3D Distill) + masks | ScanNet20 | 1.8 | 2.3 | 2.7 | 10.1 | 13.4 | 15.4 | 4.1 | 0.4 |
| OpenScene [52] (2D/3D Ens.) + masks | ScanNet20 | 2.4 | 2.8 | 3.3 | 10.4 | 13.7 | 16.3 | 4.6 | 0.9 |
| OpenMask3D (Ours) | ScanNet20 | **11.9** | **15.2** | **17.8** | **14.3** | **18.3** | **21.2** | **12.6** | **11.5** |
| OpenMask3D (Ours) | ScanNet200 | 15.0 | 19.7 | 23.1 | 16.2 | 20.6 | 23.1 | 15.4 | 14.9 |

**Table 3: 3D instance segmentation results using masks from mask module trained on ScanNet20 annotations, evaluated on the ScanNet200 dataset [57].** We identify 53 classes (such as chair, folded chair, table, dining table ...) that are semantically close to the original ScanNet20 classes [10], and group them as "Base". Remaining 147 classes are grouped as "Novel". We also report results on the full set of labels, titled "All".

**Ablation study.** In Tab. 4, we analyze design choices for OpenMask3D, *i.e.*, *multi-scale cropping* and *2D mask segmentation*. 2D mask segmentation refers to whether we use SAM [36] for refining

| 2D Mask | Multi-Scale | AP | $AP_{50}$ | $AP_{25}$ | head(AP) | common(AP) | tail(AP) |
|---------|-------------|------|------|------|----------|------------|----------|
| ✗ | ✗ | 12.9 | 16.6 | 19.5 | 15.1 | 12.2 | 11.3 |
| ✗ | ✓ | 14.3 | 18.4 | 21.1 | 16.1 | 13.6 | 12.9 |
| ✓ | ✗ | 14.1 | 18.3 | 21.5 | 16.0 | 13.0 | 13.4 |
| ✓ | ✓ | **15.4** | **19.9** | **23.1** | **17.1** | **14.1** | **14.9** |

**Table 4: OpenMask3D Ablation Study.** *2D mask* and *multi-scale crop* components. *2D mask* refers to whether SAM [36] was employed for computing 2D masks. Results are reported on the ScanNet200 [10] validation set.

| Model | Oracle | Img. Feat. | AP | head(AP) | common(AP) | tail(AP) |
|-------|--------|------------|------|----------|------------|----------|
| *Closed-vocabulary, full sup.* | | | | | | |
| Mask3D [58] | ✗ | – | 26.9 | 39.8 | 21.7 | 17.9 |
| Mask3D [58] | ✓ | – | 35.5 | 55.2 | 27.2 | 22.2 |
| *Open-vocabulary* | | | | | | |
| OpenScene [52] (2D Fusion) + masks | ✓ | OpenSeg [20] | 22.9 | 26.2 | 22.0 | 20.2 |
| OpenScene [52] (2D Fusion) + masks | ✓ | LSeg [43] | 11.8 | 26.9 | 5.2 | 1.7 |
| OpenMask3D (Ours) | ✓ | CLIP [55] | **29.1** | **31.1** | **24.0** | **32.9** |

**Table 5: 3D instance segmentation results on the ScanNet200 validation set, using oracle masks.** We use ground truth instance masks for computing the per-mask features. We also report results from the fully-supervised close-vocabulary Mask3D (row 2) whose predicted masks we match to the oracle masks using Hungarian matching, and assign the predicted labels to oracle masks. On the long-tail categories, OpenMask3D outperforms even the fully supervised Mask3D with oracle masks.

the 2D mask from which we obtain a 2D bounding box. When SAM is not used, we simply crop the tightest bounding box around the projected 3D points. The ablation study shows that both multi-scale cropping and segmenting 2D masks to obtain more accurate image crops for a given object instance positively affect the performance. Additional experiments analyzing the importance of the number of views (k) used in *top-k view selection* are provided in Appendix B.1.

**How well would our approach perform if we had *perfect* masks?** Another analysis we conduct is related to the class-agnostic masks. As the quality of the masks plays a key role in our process, we aim to quantify how much the performance could improve if we had "perfect" oracle masks. For this purpose, we run OpenMask3D using ground truth instance masks from the ScanNet200 dataset instead of using our predicted class-agnostic instance masks. In a similar fashion for the baselines, we use the oracle masks to aggregate OpenScene per-point features to obtain per-mask features. We first compare against the fully-supervised Mask3D performance (Tab. 5, row 1). In a second experiment, we also supply oracle masks to Mask3D (Tab. 5, row 2) to ensure a fair comparison. For this experiment, we perform Hungarian matching between the predicted masks and oracle masks discarding all class-losses, and we only match based on the masks. To each oracle mask, we assign the predicted class label of the closest-matching mask from Mask3D. In Tab. 5, it is evident that the quality of the masks plays an important role for our task. Remarkably, our feature computation approach that is *not* trained on any additional labeled data, when provided with oracle masks, even surpasses the performance of the fully supervised method Mask3D (by +15.0% AP) and Mask3D with oracle masks (by +10.7% AP) on the *long-tail* categories. Overall, this analysis indicates that our approach has indeed promising results which can be further improved by higher quality class-agnostic masks, without requiring any additional training or finetuning with labeled data.

## 4.2 Qualitative results

In Fig. 4, we share qualitative results from our approach for the open-vocabulary 3D instance segmentation task. With its zero-shot learning capabilities, OpenMask3D is able to segment a given query object that might not be present in common segmentation datasets. Furthermore, object properties such as colors, textures, situational context and affordances are successfully recognized by OpenMask3D. Additional qualitative results in this direction are provided in Appendix D.

In Fig. 5, we provide qualitative comparisons between our instance-based open-vocabulary representation and the point-based representation provided by OpenScene [52]. Our method OpenMask3D computes the similarity between the query embedding and per-mask feature vectors for each object *instance*, which results in crisp instance boundaries. This is particularly suitable for the use cases in which one needs to identify object instances. Additional analysis and further details about the visualizations are provided in Appendix C.

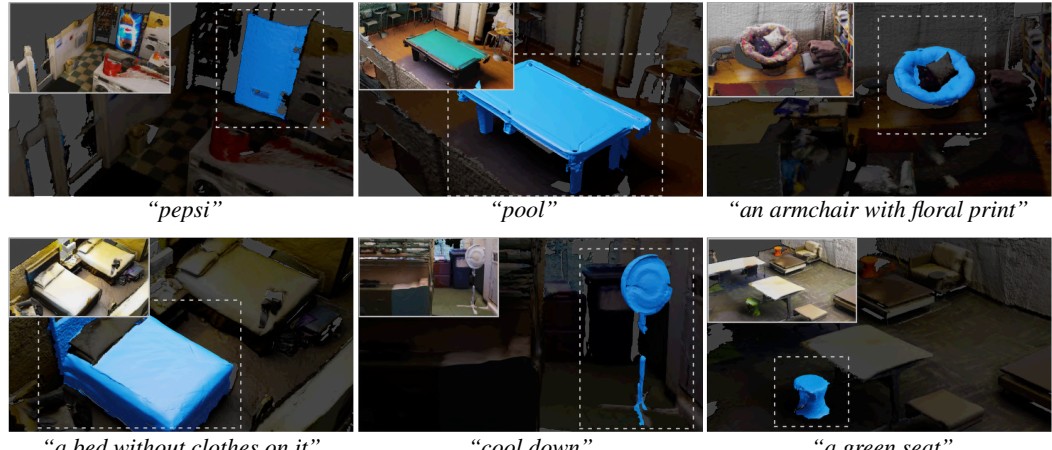

*"pepsi"*        *"pool"*        *"an armchair with floral print"*

*"a bed without clothes on it"*        *"cool down"*        *"a green seat"*

**Figure 4: Qualitative results from OpenMask3D.** Our open-vocabulary instance segmentation approach is capable of handling different types of queries. Novel object classes as well as objects described by colors, textures, situational context and affordances are successfully retrieved by OpenMask3D.

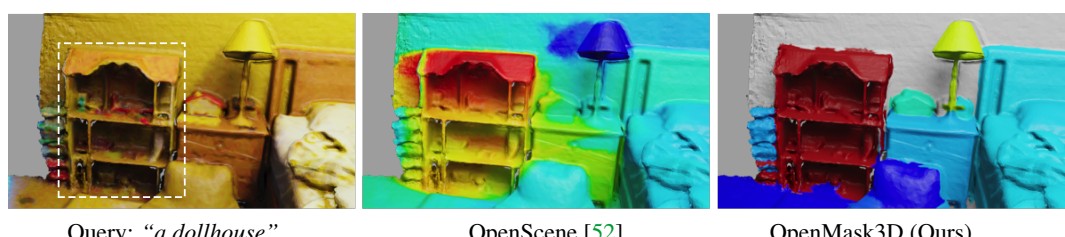

Query: *"a dollhouse"*        OpenScene [52]        OpenMask3D (Ours)

**Figure 5: Heatmaps showing the similarity between given text queries and open-vocabulary scene features.** Input 3D scene and query *(left)*, per-point similarity from OpenScene *(middle)* and per-mask similarity from OpenMask3D *(right)*. Dark red means high similarity, and dark blue means low similarity with the query text.

### 4.3 Limitations.

The experiments conducted with oracle masks indicate that there is room for improvement in terms of the quality of 3D mask proposals which will be addressed in future work. Further, since the per-mask features originate from images, they can only encode scene context visible in the camera frustum, lacking a global understanding of the complete scene and spatial relationships between all scene elements. Finally, evaluation methodologies for systematically assessing open-vocabulary capabilities still remain a challenge. Closed-vocabulary evaluations, while valuable for initial assessments, fall short in revealing the true extent of open-vocabulary potentials of proposed models.

## 5 Conclusion

We propose OpenMask3D, the first open-vocabulary 3D instance segmentation model that can identify objects instances in a 3D scene, given arbitrary text queries. This is beyond the capabilities of existing 3D semantic instance segmentation approaches, which are typically trained to predict categories from a closed vocabulary. With OpenMask3D, we push the boundaries of 3D instance segmentation. Our method is capable of segmenting object instances in a given 3D scene, guided by open-vocabulary queries describing object properties such as semantics, geometry, affordances, material properties and situational context. Thanks to its zero-shot learning capabilities, OpenMask3D is able to segment multiple instances of a given query object that might not be present in common segmentation datasets on which closed-vocabulary instance segmentation approaches are trained. This opens up new possibilities for understanding and interacting with 3D scenes in a more comprehensive and flexible manner. We encourage the research community to explore open-vocabulary approaches, where knowledge from different modalities can be seamlessly integrated into a unified and coherent space.

**Acknowledgments and disclosure of funding.** Francis Engelmann is a postdoctoral research fellow at the ETH AI Center. This project is partially funded by the ETH Career Seed Award "Towards Open-World 3D Scene Understanding", and Innosuisse grant (48727.1 IP-ICT). We sincerely thank Jonas Schult for helpful discussions, and Lorenzo Liso for the help with setting up our live demo.

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

# Appendix

The Appendix section is structured as follows: In Sec. A, we provide further details about our OpenMask3D method, and we explain certain design choices. In Sec. B we present additional results, such as ablation studies on hyperparameters and insights into the performance of OpenMask3D. In Sec. C, we provide further details about how the baseline experiments are performed. In Sec. D, we share additional qualitative results.

# A   OpenMask3D details

## A.1   Class-agnostic mask proposal module

As OpenMask3D is an instance-centric approach, we first need to obtain instance mask proposals. Since we are operating in an open-vocabulary setting, these instance masks are not associated with any class labels, they are class agnostic.

We use the mask module of the transformer-based Mask3D [58] architecture as the basis for our class-agnostic mask proposal module. Specifically, for our experiments on ScanNet200 [57], we use [58] trained on the training set of ScanNet200 [57] for instance segmentation. For our experiments on the Replica [61] dataset, we use the [58] module trained on the training and validation sets of the ScanNet200 [57] dataset, but without *segments* (see [58] for details on *segments*). We keep the weights of the mask proposal module frozen. Unlike Mask3D, our mask proposal module discards class predictions and mask confidence scores (which are based on class likelihoods), and we only retain the binary instance mask proposals.

The architecture consists of a sparse convolutional backbone based on the MinkowskiUNet [9], and a transformer decoder. Point features obtained from the feature backbone are passed through the transformer decoder, which iteratively refines the instance queries, and predicts an instance heatmap for each query. The query parameter specifies the desired number of mask proposals from the transformer-based architecture. We set the number of queries to 150, following the original implementation of Mask3D. This choice enables us to obtain a sufficient number of mask proposals for our open-vocabulary setting. The output from the final mask-module in the transformer decoder is a set of binary instance masks.

The original implementation of the Mask3D [58] model first ranks the proposed instance masks based on their confidence scores, and retains the top $k$ masks based on this ranking. As the confidence scores are guided by class likelihoods, we do not utilize such scores to rank the masks, and do not filter out any of the proposed instance masks. Furthermore, since we aim to retain as many mask proposals as possible, we do not perform mask-filtering based on object sizes, or pairwise mask overlaps. In other words, we keep all masks proposed by the mask module as each of these masks might correspond to a potential open-vocabulary query.

As the model may occasionally output mask proposals that are not spatially contiguous, the original implementation of Mask3D employs the DBSCAN clustering algorithm [17] to break down such non-contiguous masks into smaller, spatially contiguous clusters. We follow this practice, and perform DBSCAN clustering on the instance masks, setting the epsilon parameter, $\epsilon$, to $0.95$. This procedure often increases the final number of instance masks. As such, our OpenMask3D model generates class-agnostic instance masks, for which we compute open-vocabulary mask-features as described next.

## A.2   Mask-feature computation module details

In the main paper, we outlined the steps to extract per-mask features in the CLIP [55] space. In this section, we will delve into the motivations behind specific design choices made throughout the process.

### A.2.1   Per-mask view ranking: assumptions and visualizations

Once we obtain the class-agnostic masks, our primary objective is to determine the best CLIP features for each of them. We operate under the assumption that CLIP is capable of extracting meaningful features from instances when provided with a favorable viewpoint, and in our feature computation process, we prioritize selecting views that offer a higher number of visible points. In practice, to assess the quality of a particular viewpoint for each mask, we compute the proportion of points of the 3D instance mask that are visible from that view. Next, we rank the views based on their visibility scores, and we select the top $k$ views with the highest scores, where $k$ is a hyper parameter. Fig. 6 illustrates this concept, where two views of a given instance are compared based on the number of visible points. The view that has a higher number of visible points from the instance is selected (outlined in green).

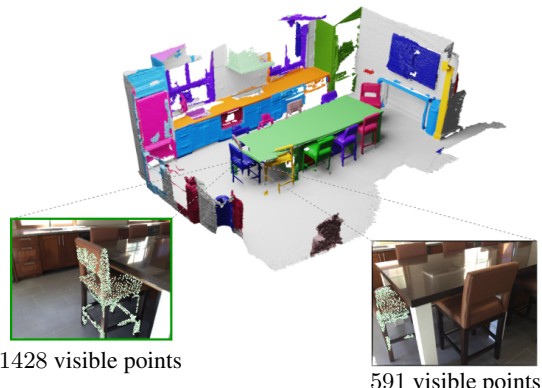

1428 visible points

591 visible points

**Figure 6:** Visual comparison between a selected view (marked with a green outline) for the corresponding chair instance, and a discarded view. The green points in the images represent the 3D instance points projected onto the 2D image. In the selected view the instance is fully visible, and mostly occluded in the discarded view.

### A.2.2 Why do we need SAM?

In order to get the CLIP image features of a particular instance in the selected views, we need to crop the image section containing the object instance. This process involves obtaining an initial tight crop of the instance projected onto 2D, and then incrementally expanding the crop to include some visual context around it.

A straightforward approach to perform the cropping would be to project all of the visible 3D points belonging to the mask onto the 2D image, and fit a 2D bounding box around these points. However, as we do not discard any mask proposals, the masks can potentially include outliers, or the masks might be too small as an outcome of the DBSCAN algorithm described in Sec. A.1. As depicted in Fig. 7, this can result in bounding boxes that are either too large or too small. Using CLIP-image features obtained from crops based solely on projected points can lead to inferior instance mask-features.

To address this challenge, and to improve the robustness of our model against noisy instance masks, we propose using a 2D segmentation method that takes a set of points from the mask as input, and produces the corresponding 2D mask, together with its confidence score. The Segment Anything Model (SAM) [36] precisely fulfills this task, allowing us to generate accurate 2D masks in a class-agnostic manner.

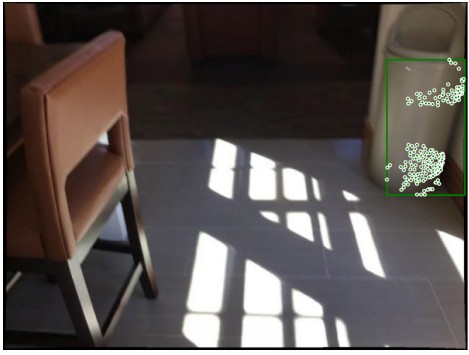
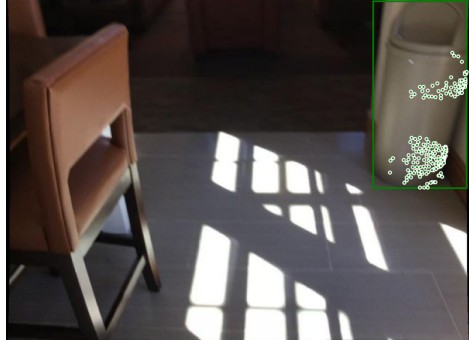

**Figure 7:** Difference between the bounding boxes obtained by tightly cropping around the projected points from the 3D instance mask (left), and the bounding box obtained from the 2D mask generated by SAM (right).

### A.2.3 Which points should we give as input to SAM?

When using SAM for predicting a 2D mask, we need to determine a set of points on which the mask generation process will be conditioned. Initially, we attempted to input all the visible points projected from the 3D mask. However, this approach results in poor quality masks due to the inaccuracies of the projections and the noise present in the 3D masks, as illustrated in Figure 8.

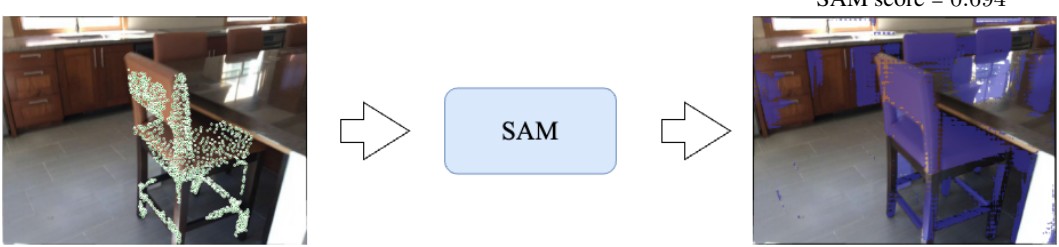

SAM score = 0.694

**Figure 8:** Output of SAM, using all of the visible points from the projected 3D mask as input.

To address this issue, we explore an alternative approach. Instead of using all of the visible points as input for SAM, we randomly sample a subset of points from the projected 3D mask. Interestingly, this method produces much cleaner masks, as it can be seen in Fig. 9.

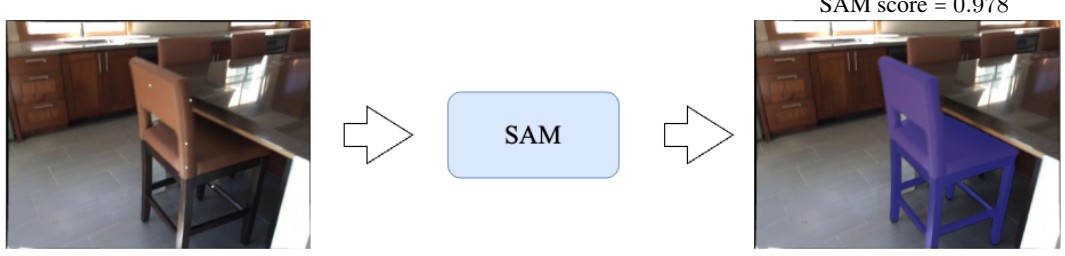

SAM score = 0.978

**Figure 9:** Output of SAM, using only 5 randomly sampled points (visualized as green dots) of the projected 3D mask as input.

### A.2.4 Why do we need to run SAM for multiple rounds?

Relying solely on a small set of random points as input for SAM might be unreliable, particularly when the sampled points are outliers, or too concentrated in a particular area of the instance we want to segment.

To address this limitation, we implemented a sampling algorithm (as outlined in the main paper) inspired by RANSAC [18]. In this approach, we perform $k_{round}$ = 10 sampling rounds, and in each round we randomly sample $k_{sample}$ = 5 points from the projected 3D instance mask. SAM returns the predicted 2D mask along with the corresponding mask confidence score for each set of sampled points. Based on the confidence scores returned by SAM, we select the mask with the highest score. Fig. 10 illustrates an example where the points sampled in one round are concentrated in a small spatial range, resulting in an incorrect mask prediction by SAM.

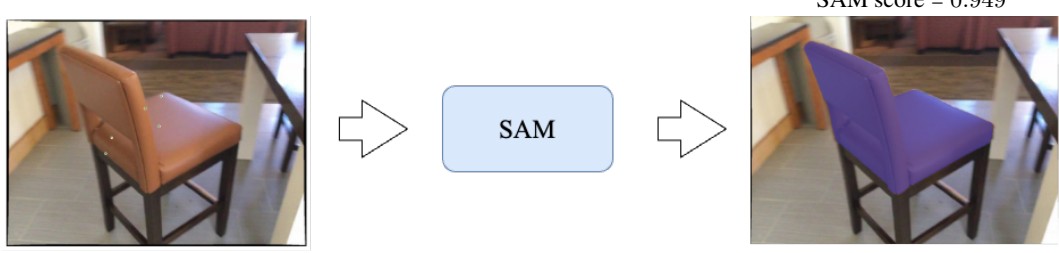

SAM score = 0.949

**Figure 10:** Output of SAM, using only 5 randomly sampled points of the mask as input. Here the sampled points (the green points visualized in the image) are concentrated in a small spatial range and cause SAM to predict an incorrect mask, which does not include the legs of the chair.

However, by employing our sampling algorithm and running SAM for multiple rounds, we achieve improved results. The iterative process allows us to select the mask with the highest score achieved among the rounds, as shown in Fig. 11. In this particular example, the selected mask, i.e. the one with the highest SAM confidence score, accurately segments the chair.

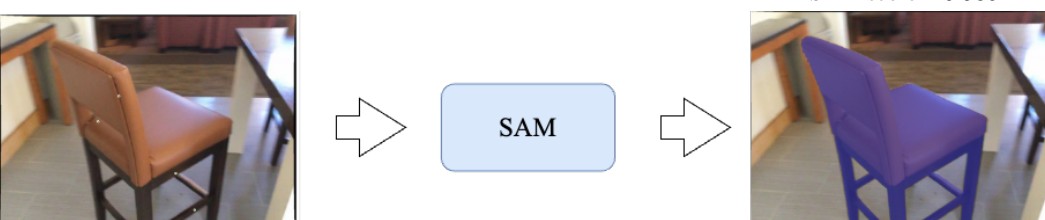

SAM score = 0.985

**Figure 11:** Illustration of the results from the round which gives the highest confidence score. On the left, we visualize the 5 sampled points which are given as input to SAM. On the right, we visualize the SAM mask prediction, showing a perfect segmentation of the chair.

# B   Additional results

## B.1   Ablation Studies

In addition to our ablation experiments provided in the main paper, here we share additional results. First, in Table 6, we share results obtained from using a varying number of views ($k$) in the top-$k$ view selection process. This analysis is conducted on the ScanNet200 dataset. Additionally, we provide an ablation study evaluating the hyperparameters of the multi-scale cropping in our method. This analysis is conducted on the Replica dataset, and is presented in Table 7.

| Top-k | AP | $AP_{50}$ | $AP_{25}$ | head(AP) | common(AP) | tail(AP) |
|---|---|---|---|---|---|---|
| 1 | 13.6 | 17.6 | 20.8 | 15.5 | 12.5 | 12.8 |
| 5 | **15.4** | 19.9 | 23.1 | **17.1** | **14.1** | 14.9 |
| 10 | **15.4** | **20.0** | **23.2** | 16.4 | 14.0 | **15.8** |

**Table 6:  Ablation study of the top-$k$ frame selection parameter $k$.**  This analysis is conducted on the ScanNet200 validation set.

| Levels | Ratio of Exp. | AP | $AP_{50}$ | $AP_{25}$ |
|---|---|---|---|---|
| 1 | 0.1 | 11.3 | 16.0 | 20.2 |
| 3 | 0.1 | **13.1** | **18.4** | **24.2** |
| 5 | 0.1 | 12.8 | 17.6 | 22.6 |
| 3 | 0.05 | 12.9 | 18.1 | 23.5 |
| 3 | 0.1 | **13.1** | **18.4** | **24.2** |
| 3 | 0.2 | 12.8 | 17.7 | 22.9 |

**Table 7: Ablation study of the multi-scale cropping hyperparameters on the Replica dataset.** We analyze the effect of varying number of levels, and the ratio of expansion.

## B.2   Evaluation on Replica without RGB-D images

Our approach OpenMask3D requires images as input, as it depends on visual-language models that operate on images in combination with text. In this work, we prioritized the ability to recognize uncommon/long-tail objects over generalization across different modalities. Using vision-language models on images provides an excellent opportunity to preserve this generalization capability. Nevertheless, even when only the 3D scan of a scene is available, it could still be possible to render images from the 3D scan, and use those synthetic images as input to our OpenMask3D pipeline. We tried this on the Replica [61] dataset, and rendered RGB-D images from the dense scene point clouds (as illustrated in Fig. 12). Our results using these rendered synthetic views of the scene point cloud are presented in Tab. 8. Even without any RGB-D images as an input, OpenMask3D performs relatively well, and we observe that the performance decreases by only −1.5 AP. Furthermore, OpenMask3D shows stronger performance compared to the 3D baseline (OpenScene [52], 3D Distill).

# C   Details on baseline experiments

## Quantitative baseline experiments

As mentioned in the main paper, to compare our OpenMask3D method with the 3D open-vocabulary scene understanding approach OpenScene [52], we adapted OpenScene for the 3D instance segmentation task. As

| Model | AP | AP$_{50}$ | AP$_{25}$ |
|---|---|---|---|
| OpenScene [52] (3D Distill) + masks | 8.2 | 10.5 | 12.6 |
| OpenMask3D w. rendered RGB-D images | 11.6 | 14.9 | 18.4 |
| OpenMask3D | **13.1** | **18.4** | **24.2** |

**Table 8: 3D instance segmentation results on the Replica dataset.** Comparing OpenMask3D performance when using original RGB-D images vs. RGB-D images rendered from the point cloud.

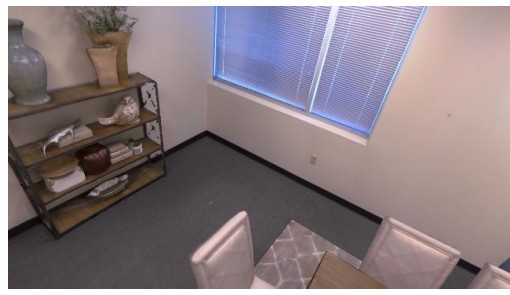

**(a)** RGB image from the Replica dataset

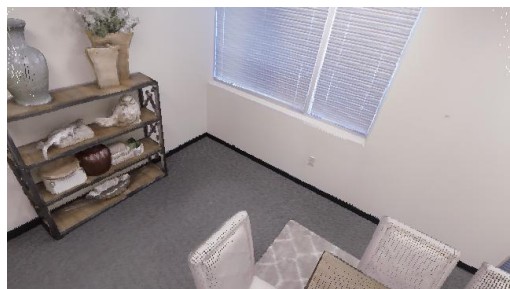

**(b)** RGB image rendered from the point cloud

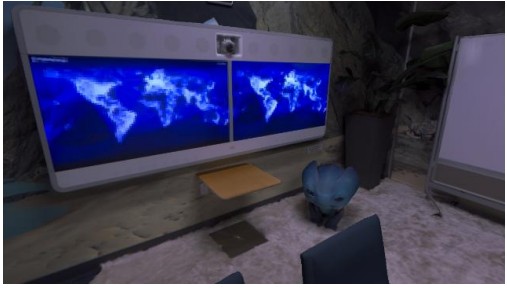

**(c)** RGB image from the Replica dataset

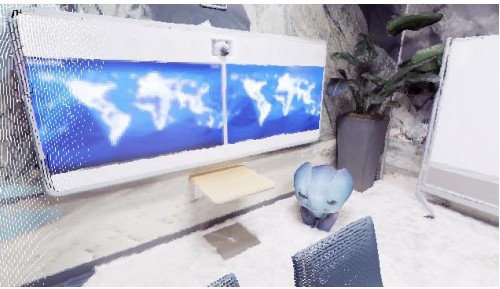

**(d)** RGB image rendered from the point cloud

**Figure 12: Visualization of the Replica RGB images.** Original RGB images from the Replica dataset (left), and RGB images rendered from the scene point cloud (right).

OpenScene computes a per-point feature representation for each point in the point cloud, we use our proposed class-agnostic instance masks to aggregate OpenScene features for each mask. This way, we can associate each object instance mask with a "per-mask" feature computed from OpenScene features.

OpenScene provides 3 different model variants, namely 2D fusion, 3D distill, and 2D/3D ensemble. We primarily compare with the OpenScene models using OpenSeg [20] features, whereas we also experiment with the 2D fusion model using LSeg [43] features. The main reason why we primarily compare against the models using OpenSeg features is that the features have a dimensionality of 768, relying on the same CLIP architecture we employ in our work – ViT-L/14 (336px) [55] –, whereas the LSeg features have a dimensionality of 512. For our closed-vocabulary experiments with the OpenScene model using LSeg features, we embed each text category using ViT-B/32 CLIP text encoder, and we assign object categories using these embeddings. In all other experiments, we embed text queries using the CLIP architecture ViT-L/14 (336px).

## Qualitative baseline experiments

In addition to the quantitative results presented in the main paper, we conducted qualitative comparisons to further evaluate our mask-based open-world representation in comparison to the point-based representation provided by OpenScene [52].

In Fig. 13 and Fig. 14, we provide these qualitative comparisons. In Fig. 13, we show results from the OpenScene [52] 2D fusion model using OpenSeg [20] features, and from OpenMask3D. In Fig. 14, we compare features from OpenMask3D with features from several OpenScene variants (2D fusion, 3D distill, 2D/3D ensemble). To visualize results from OpenScene, for a given query, we compute the similarity score between each point-feature and the query text embedding, and visualize the similarity scores for each point as a heatmap over the points. On the other hand, OpenMask3D results are based on mask-features, i.e., each mask has an associated feature vector, for which we compute a similarity score. Hence, we visualize the similarity scores for each mask as a heatmap over the instance masks, and each point in a given instance mask is assigned to the same similarity score.

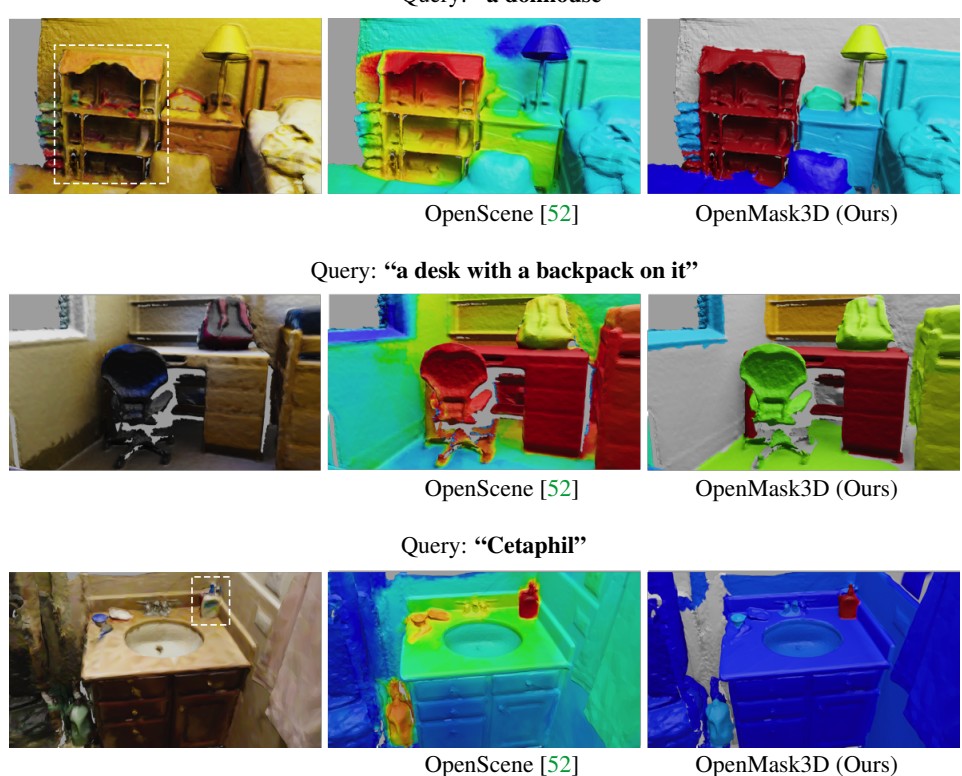

**Figure 13: Qualitative comparisons.** Heatmaps illustrating the similarity between the CLIP embeddings of a specific query and the open-vocabulary representation of the scene. A comparison is made between the point-based OpenScene approach proposed by Peng et al. [52] (left) and our mask-based approach OpenMask3D (right). Dark red means high similarity, and dark blue means low similarity with the query.

As evident from Fig. 13 and Fig. 14, while similarity computation between the query embedding and each OpenScene per-point feature vector results in a reasonable heatmap, it is challenging to extract object instance information from the heatmap representation. Our method OpenMask3D, on the other hand, computes the similarity between the query embedding and each per-mask feature vector. This results in crisp instance boundaries, particularly suitable for the use cases in which one needs to handle object instances.

# D Qualitative results

In this section, we provide additional qualitative results.

In Fig. 15, we show results from object categories that are not present in the ScanNet200 label set. Objects from these novel categories are successfully segmented using our OpenMask3D approach. Thanks to its zero-shot learning capabilities, OpenMask3D is able to segment a given query object that might not be present in common segmentation datasets on which closed-vocabulary instance segmentation approaches are trained.

In Fig. 16, we show open-vocabulary 3D instance segmentation results using queries describing various object properties such as affordances, color, geometry, material type and object state. This highlights that our model is able to preserve information about such object properties and go beyond object semantics, contrary to the capabilities of existing closed-vocabulary 3D instance segmentation approaches.

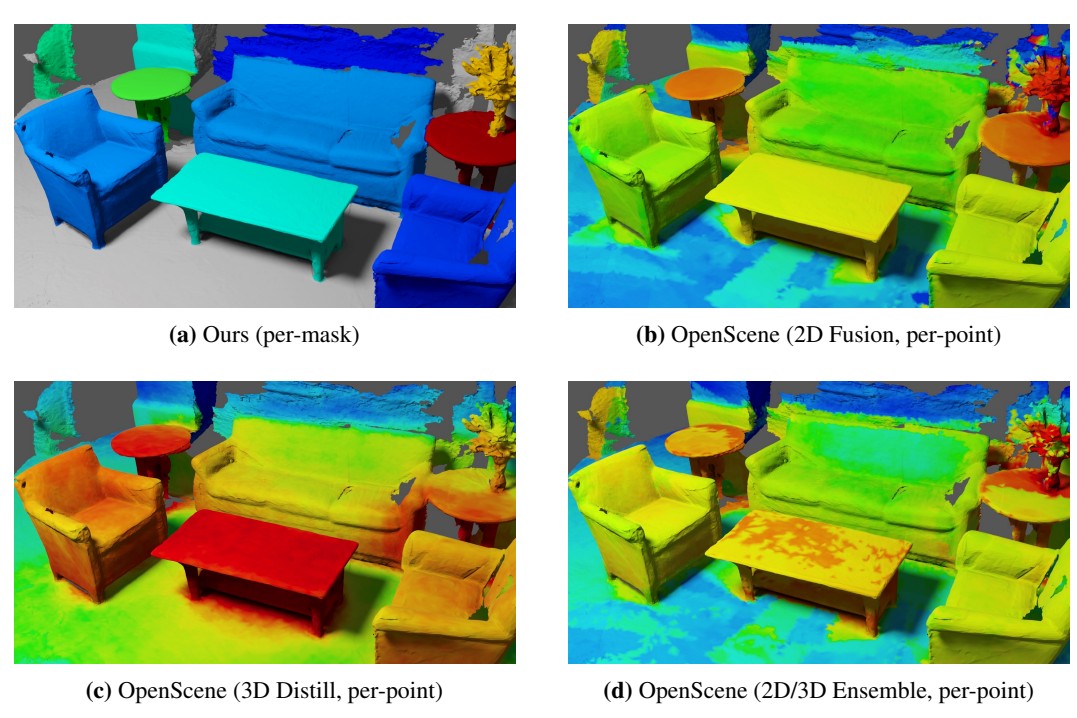

**(a)** Ours (per-mask)

**(b)** OpenScene (2D Fusion, per-point)

**(c)** OpenScene (3D Distill, per-point)

**(d)** OpenScene (2D/3D Ensemble, per-point)

**Figure 14:** Feature similarity for the query "the table with a flower vase on it".

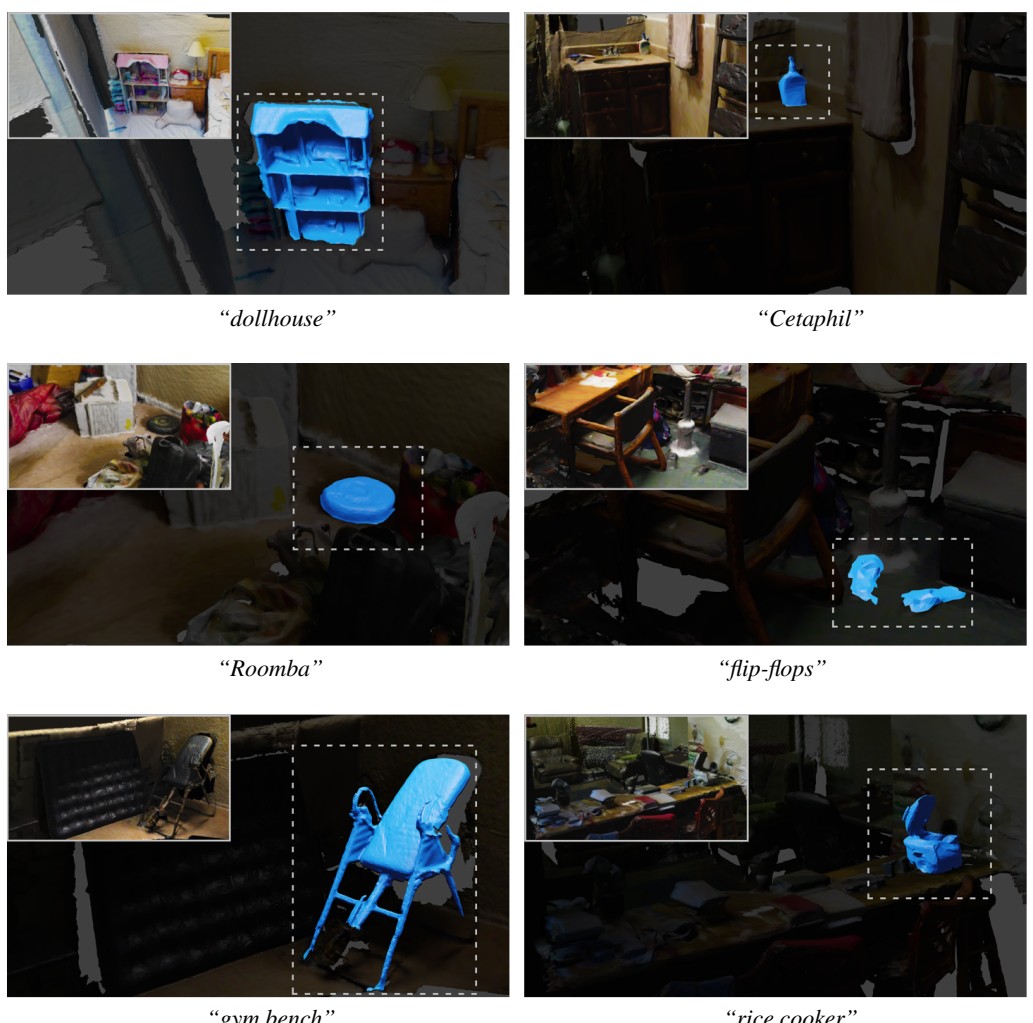

*"dollhouse"*            *"Cetaphil"*

*"Roomba"*            *"flip-flops"*

*"gym bench"*            *"rice cooker"*

**Figure 15: Qualitative results from OpenMask3D.** We show open-vocabulary instance segmentation results using arbitrary queries involving object categories that are not present in the ScanNet200 dataset labels. In each scene, we visualize the instance with the highest similarity score for the given query embedding. These predictions show the zero-shot learning ability of our model, highlighting the open-vocabulary capabilities.

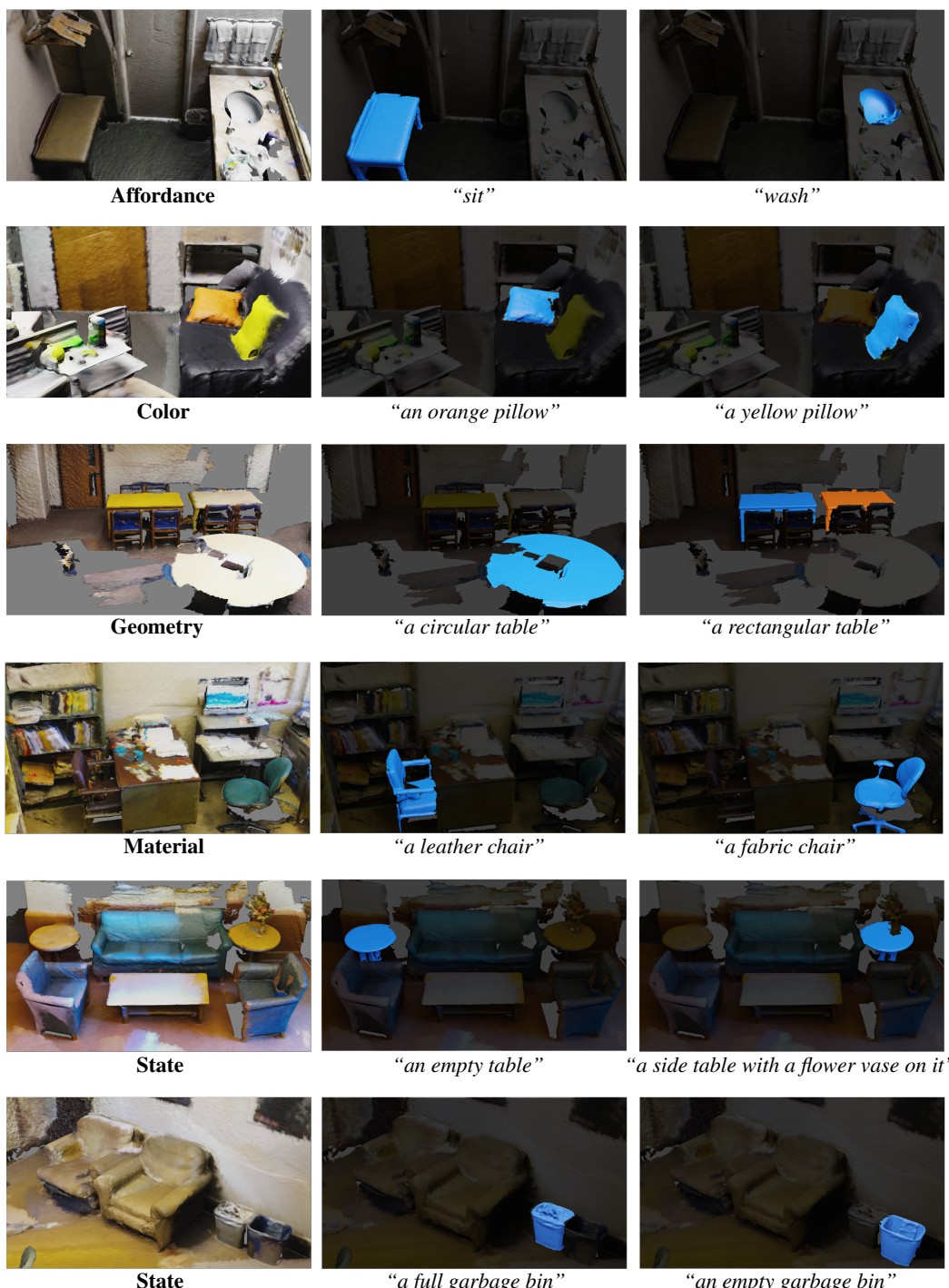

**Figure 16: Qualitative results from OpenMask3D, queries related to object properties.** We show open-vocabulary 3D instance segmentation results using queries describing various object properties such as affordances, color, geometry, material type, and object state. In each row, we show two different queries per category, for which our method successfully segments different object instances. These results highlight the strong open-vocabulary 3D instance segmentation capabilities of our model, going beyond object semantics.

