# OpenReview forum: "OpenMask3D: Open-Vocabulary 3D Instance Segmentation"
_NeurIPS.cc/2023/Conference — NeurIPS 2023 poster_

### Official Review · Reviewer_STxk · 2023-06-23

**Soundness:** 3 good
**Presentation:** 3 good
**Contribution:** 2 fair
**Rating:** 5
**Confidence:** 3

**Summary:**

This paper focuses on the task of open-vocabulary 3D instance segmentation, which involves predicting 3D object instance masks and their corresponding categories. The authors highlight the limitations of traditional closed-vocabulary approaches that operate within a predefined set of object categories, which restricts their ability to handle novel objects and free-form queries.

To address these limitations, the authors propose OpenMask3D, a zero-shot approach for open-vocabulary 3D instance segmentation. OpenMask3D utilizes predicted class-agnostic 3D instance masks and performs mask-feature aggregation using CLIP-based image embeddings. This allows the model to reason beyond pre-defined concepts and handle open-vocabulary queries.

This paper demonstrates its superiority over closed-vocabulary counterparts. The proposed method has the potential to enhance various applications such as robotics, augmented reality, scene understanding, and 3D visual search.

**Strengths:**

The authors propose OpenMask3D as the first zero-shot approach for open-world 3D instance segmentation, offering a unique solution that leverages predicted class-agnostic 3D instance masks and CLIP-based image embeddings. This combination of ideas and techniques demonstrates originality in problem formulation, model architecture, and feature aggregation for open-vocabulary 3D instance segmentation.

The methodology is presented in a clear and structured manner, allowing readers to understand the proposed approach easily. The experimental evaluation is conducted on the common benchmark datasets, and the authors perform ablation studies to gain insights into the design choices of the model.

The results demonstrate that OpenMask3D outperforms other open-vocabulary counterparts, particularly in scenarios with a long-tail distribution of objects, which advance the state-of-the-art 3D instance segmentation in the complex real world.


**Weaknesses:**

While the paper demonstrates several strengths, there are also some areas that could be improved to further enhance its contributions and impact.

Evaluation on a Single Dataset: The paper conducts experiments and ablation studies on the ScanNet200 dataset, which may limit the generalizability of the findings. It would be beneficial to evaluate the proposed OpenMask3D method on multiple datasets, including those with varying object distributions, to assess its robustness and performance across different scenarios. A more diverse evaluation would strengthen the claims made in the paper.

Long Pipeline: another weakness of this work is the long pipeline of the proposed OpenMask3D model. The pipeline consists of multiple stages, including a class-agnostic mask proposal head, mask-feature aggregation module, and multi-view fusion of CLIP-based image embeddings. The length of the pipeline may introduce computational complexity and potentially impact the efficiency and real-time applicability of the model.

Lack of Computation Cost Comparison: While the paper discusses the architecture and methodology of OpenMask3D in detail, it does not provide a thorough comparison of the computational resources required and the speed achieved between different methods.

Problematic Structure of section 4.1: sec 4.1 is named as "Closed-vocabulary 3D instance segmentation evaluation", while both closed-set instance segmentation results and open-vocabulary instance segmentation results are presented in this section.


**Questions:**

The paper primarily evaluates the proposed OpenMask3D method on the ScanNet200 dataset. Evaluation on more datasets with diverse scenes should be performed.

It would be beneficial to include a comprehensive comparison of the computational resources and inference speed between OpenMask3D and existing closed-vocabulary and open-vocabulary 3D instance segmentation methods.

The structure of Experiment section should be better organized.



**Limitations:**

The paper briefly mentions limitations of  OpenMask3D. It would be valuable to provide a more thorough analysis of the limitations, and discuss potential method / direction that could be explored to solve it.

Potential negative societal impact is not applicable for this work.

---

> ### Author Rebuttal · Authors · 2023-08-09
>
> We thank the reviewer for the valuable feedback, and aim to address the questions in the following:
>
> ### Q1: Evaluation on more datasets and settings
>
> The reviewer suggests evaluating on other datasets to show the generalizability of OpenMask3D.
> We agree and additionally run experiments on Replica, besides ScanNet200.
> For the OpenScene baselines, we use the pre-computed fused features from the OpenScene github repository. Scores are shown in the table below. The results show that our approach generalizes well to new datasets, we obtain comparable scores as on ScanNet200.
>
> | Method | ScanNet200 (mAP$_{50}$)  | Replica (mAP$_{50}$|
> | --- | --- | --- |
> | OpenScene (2D Fusion) | 15.2 | 15.6 |
> | OpenScene (3D Distill) | 6.2 | 10.5 |
> | OpenScene (2D/3D Ens.) | 6.7 | 10.4 |
> | OpenMask3D (Ours) | 16.8 | 18.4 |
>
> We additionally provide an experiment to analyze the generalization capability of our approach beyond the training categories. We use class-agnostic masks from a mask-predictor trained on the 20 original ScanNet classes, and evaluate on the ScanNet200 dataset. As we intend to assess how well our model performs on “unseen” categories, we categorize objects into two subsets: *base* and *novel* classes. We identify ScanNet200 categories that are semantically close to the original ScanNet20 classes (folded-chair, table, dining table, cabinet, kitchen cabinet, bathroom counter etc.), resulting in 53 categories. Below, we report results on seen (“base”) classes, unseen (“novel”) classes, and all classes. Full table is provided in Tab. 1 of the rebuttal PDF.
>
> |Method|Novel (AP$_{50}$)|Base (AP$_{50}$)|All (AP$_{50}$)|
> |---|---|---|---|
> | OpenScene (2D Fusion) |10.3|**15.0**|11.6|
> | OpenScene (3D Distill) |2.3|13.4|5.3|
> | OpenScene (2D/3D Ens.) | 2.8| 13.7|5.8|
> | OpenMask3D (Ours) |**12.9**| **15.0**| **13.5**|
>
> ### Q2: Comparison of Computational Resources
>
> We compare the necessary computational resources of the original OpenScene from [46] and our OpenMask3D. We report the runtimes and GPU memory requirements.
>
> ****1) Runtime****
>
> In the table below, we provide the runtime of our approach on ScanNet for generating the queryable 3D scene representation, and also for performing a query on that representation. For comparison, we additionally provide the runtimes of the original OpenScene [46]. All runtimes are measured on the same hardware.
> | Method | Generating 3D Scene Representation [s] | Querying 3D Scene Representation [ms] | Semantics? | Instances? |
> | --- | --- | --- | --- | --- |
> | OpenScene (2D Fusion) |  440 | 168.8 | $\checkmark$| ✘  |
> | OpenScene (2D3D Ensemble) |  524 | 168.8 | $\checkmark$| ✘  |
> | OpenMask3D |  556 | **0.923** |  $\checkmark$| $\checkmark$ |
> | OpenMask3D (fast) |  **350** | **0.923** |  $\checkmark$| $\checkmark$ |
>
> We also report the runtime of a faster variation of our model (called ‘fast’) which runs CLIP only once per segment on a single crop and uses a smaller SAM backbone. The exact hyper-parameters are shown in the table below. This faster version performs almost as good as the original model (small performance drop of -1.3 mAP$_{50}$).
>
> | Model | top k | $\mathsf{L}$ | SAM backbone |
> | --- | --- | --- | --- |
> | OpenMask3D |  5 | 3 |  ViT-H|
> | OpenMask3D (fast) |  1 | 1 |  ViT-B|
>
>
> ****2) Minimum GPU Memory Requirement to run the Model****
>
> In the table below we indicate the minimum GPU memory requirements for OpenScene and OpenMask3D. OpenScene has much stronger GPU requirements since it depends on OpenSeg.
>
> | Method | GPU memory | main components |
> |---|---|---|
> | OpenScene |  32 GB | OpenSeg: 32GB, CLIP-text: 4GB, OpenScene: 32GB|
> | OpenMask3D | 10 GB | SAM: 8GB, CLIP-image: 4GB, CLIP-text:  4GB |
> | OpenMask3D-fast | 4 GB | SAM: 4GB, CLIP-image: 4GB, CLIP-text:  4GB|
>
> ### Q3: Structure of Section 4.1
>
> The reviewer highlights the interleaved presentation of both open-set and closed-set results in this section.
> This is correct and we will separate both settings more clearly to improve the organization of the experiment section.
>
> ### Limitations
> We agree with the reviewer that our paper would benefit from discussing the limitations further, and we will extend our section (L320-323) to include a discussion on class-agnostic mask quality as also discussed in (L294-308). Another limitation we would like to mention is that current top-k view selection algorithm directly relies on the visibility of an object instance in each frame - and this can benefit from additional criteria to assess the *quality* of a frame so that more informative frames could be selected. We have observed that frame-selection plays an important role, particularly for avoiding scene context to be overly infused into our per-mask features. Lastly, we also agree that a systematic approach is needed to evaluate open-vocabulary performance quantitatively, and call for future work in this direction.

---

> > ### Comment · Reviewer_STxk · 2023-08-18
> >
> > The response solves my concern, I keep my rating of 5.

---

### Official Review · Reviewer_7yua · 2023-07-01

**Soundness:** 3 good
**Presentation:** 3 good
**Contribution:** 2 fair
**Rating:** 6
**Confidence:** 4

**Summary:**

The paper proposes to perform open-vocabulary instance segmentation by utilizing class-agnostic 3D instance segmentation masks from a 3D instance segmentation model trained on scannet200 and generating class labels for it using CLIP. The paper proposed to first obtain class-agnostic instance masks from a supervised Mask3D model (without using class annotations). Then it finds the images where the objects are best visible. It then projects the 3D segmentation masks in those views and refine them using SAM. Finally they average features for each object from multiple views and at multiple scale to arrive at one feature vector per 3D instance mask. The class label can then be obtained by doing dot-product with the obtained feature vector and the language embedding of the class. The results show that the proposed method is superior than prior point-based methods when also supplied with class-agnostic instance segmentation mask in closed-vocabulary setting.  The paper also shows some qualitative results with free-flowing natural language.

**Strengths:**

- The paper is well written, I especially appreciate the helpful supplementary video and text content which made the nitty gritty details of the pipeline very clear.
- Open-Vocabulary 3D instance segmentation is a very useful task which hasn't been tackled before -- this paper brings attention to it (that said, I have some concerns here as mentioned in weaknesses)
- The proposed method obtains better results than prior point-based methods like OpenScene

**Weaknesses:**

- Open-Vocabulary: The proposed method relies on 3D instance segmentation predictions which in turn relies on 3D segmentation annotations. This, however, is not available for wide variety of objects, hence I am unsure if we can conclude that this model is indeed “open-vocabulary”. For example: If instead of scannet200, the proposed model uses class-agnostic masks from a model trained on 20 scannet classes, would it be able to achieve decent results on Scannet200? Would this model trained on scannet work on a different dataset like MatterPort3D? Additionally, since 3D datasets are significantly smaller than their 2D counterparts, doesn’t relying on 3D instance segmentation mask a serious bottleneck which wouldn’t scale? The point-based models are open-vocabulary in the sense that they are not bottlenecked by any 3D-specific annotations — at the same time I do agree with the point of this paper that they can only do semantic and not instance segmentation. However, using instance masks via 3D annotations might not as well lead to “open-vocabulary instance segmentation” proposed in this paper. Ideally, it should primarily have results on held out 3D categories that the model or any of its components have never seen during training.
- In continuation of the above, the results for 3D instance segmentation in open-vocabulary setting is only qualitative and not quantitative. I understand though that prior methods too show only qualitative open-vocabulary results, and so this is said as a minor point and not a major complaint. However, the lack of quantitative results on categories outside the training data is concerning — especially since this proposed model particularly used labels from scannet which may not generalize beyond the 200 categories they were trained on, and on out-of-domain 3D scenes.
- Unfounded Claims:
    - L304: The paper highlights that when given access to oracle masks at “test time” to their model, it outperforms the supervised Mask3D model on tail AP by 9.1%. As the paper concludes in L305-308, this result indicates that if somehow we are able to obtain high quality class-agnostic masks, we do not need supervision for class labelling as their method can outperform supervised Mask3D. In my opinion, this is a misleading claim because while their baseline “Mask3D” has access to ground truth masks and classes during training, it does not have access to oracle masks during test time. This makes the comparison unfair. AP is very sensitive to quality of the segmentation mask and if supplied oracle masks to Mask3D it may do much better. A very crude way to supply that would be — computing Hungarian matching between the predicted masks and ground truth masks (without class labels), and for each predicted mask replace it with the matched ground truth mask while keeping the class label same. In general though, this comparison of supervised mask3d and the proposed method needs much more care to make balanced conclusions.
    - (Minor) L290-291: Claims that “the ablation study show that effect of 2D mask segmentation is less significant than the effect of multi-scale cropping”. Based on this, one might expect the row 2 of component analysis section to be (significantly) better than row 3. However, on some metrics row 2 wins while on others row 3 wins. And the difference is not that much.

**Questions:**

- As mentioned in the limitations, here are some suggestions which might help alleviate the concerns over instance-segmentation annotation bottleneck:
    -  Does class-agnostic segmentation generalize beyond training categories? Performance on OOD dataset like Matterport3D and maybe instance segmentation model trained on scannet 18 classes and tested on scannet 200 class could help (compared against point-based methods). Another suggestion could be to hold out rare categories from Scannet200 for training class-agnostic instance segmentation model and evaluate on the held out categories at test time. Ofcourse, these are just suggestions and any other experiment that could help us get to the bottom of this will be highly appreciated.

- Answers to "unfounded claims" as described in limitations would be super helpful too.

- Could you give some insights on why is the performance of OpenScene 2D Fusion model so much better than 2D-3D ensemble models while the Openscene paper consistently showed better results with the ensemble version of their model? That was a bit strange to me.

**Limitations:**

The paper do not discuss limitations. I think the biggest one is their reliance on class-agnostic instance segmentation mask annotations which would be good to discuss in the ppaer.

---

> ### Author Rebuttal · Authors · 2023-08-09
>
> We thank the reviewer for the extensive feedback, we really appreciate the helpful suggestions for experimental setups!
>
> ### Open-vocabulary Evaluation & Generalization Beyond Training Categories
> The reviewer correctly highlights that Mask3D is trained on closed-set segmentation masks dataset, and is concerned that the class-agnostic Mask3D would not generalize beyond the masks seen during training. This is indeed a valid concern, and deserves further analysis. However -- as our additional experiments below indicate -- in practice, the model still manages to generalize quite well beyond the object masks seen during training. To demonstrate the capability of our approach beyond the mask-predictor training categories, we conducted a series of experiments following the suggestions by the reviewer.
>
> **Generalization to unseen categories**
>
> First, we analyze how well our model would perform if we use class-agnostic masks from a mask-predictor trained on the 20 original ScanNet classes, and evaluate on the ScanNet200 dataset. To evaluate how well our model performs on “unseen” categories, we classify the ScanNet200 labels into two subsets: *base* and *novel* classes. We identify ScanNet200 categories that are semantically similar to the original ScanNet20 classes (e.g. chair and folded-chair, table and dining-table, etc.), resulting in 53 classes. We group all remaining object classes that are not similar to any class in ScanNet20 as "novel". Below, we report results on seen (“base”) classes, unseen (“novel”) classes, and all classes. The full table is provided in Tab. 1 of the rebuttal PDF.
>
> |Method|Novel (AP$_{50}$)|Base (AP$_{50}$)|All (AP$_{50}$)|
> |---|---|---|---|
> | OpenScene (2D Fusion) |10.3|**15.0**|11.6|
> | OpenScene (3D Distill) |2.3|13.4|5.3|
> | OpenScene (2D/3D Ens.) | 2.8| 13.7|5.8|
> | OpenMask3D (Ours) |**12.9**| **15.0**| **13.5**|
>
> Our experiments show that the model trained on a smaller set of object annotations from ScanNet20 can generalize to predict object masks for a significantly larger set of objects (ScanNet200), resulting in only a marginal decrease in the performance. Particularly, we see that OpenMask3D, compared to other open-vocabulary counterparts - seem to better preserve information about uncommon objects.
>
> **Generalization to OOD data**
>
> Furthermore, we show results on out-of-distribution data from *Replica*, using a mask predictor trained on ScanNet.
> For the OpenScene baselines, we use the pre-computed features from the OpenScene repository.
> Replica dataset contains high-quality *mesh* reconstruction of indoor scenes, and RGB-D images rendered from these meshes.
> In order to assess the robustness of our CLIP-based mask-feature module to image-quality and realism, we conduct a second experiment where we render RGB-D images from the *point clouds* of Replica scenes (marked “rendered RGB-D” below, illustrated in Fig. 1.b, rebuttal PDF).
>
> |Method|AP|AP$_{50}$|AP$_{25}$|
> |---|---|---|---|
> |OpenScene (2D Fusion)|10.9|15.6|17.3|
> |OpenScene (3D Distill) |8.2|10.5|12.6|
> |OpenScene (2D/3D Ens.)|8.2|10.4|13.3|
> |OpenMask3D (rendered RGB-D) |11.6|14.9|18.4|
> | OpenMask3D |**13.1**|**18.4**|**24.2**|
>
> Our results demonstrate that OpenMask3D can indeed generalize to unseen categories as well as OOD data.
> Nevertheless, we understand and agree with the reviewer’s concern in general, however, the above experiments seem to indicate that this is less of a problem than one might initially think. In particular, the mask predictor module trained on a smaller set of objects seems to perform reasonably well in various settings. Furthermore, several qualitative examples we provide in our submission show that our method can achieve good “open-vocabulary” results for objects that were not originally annotated e.g. Fig.1 "angel", Fig.4 "pepsi" - and more in the supplementary.
>
> ### Oracle Mask Experiment
>
> The question is about an experiment from the paper (Tab. 3) showing that OpenMask3D when given access to oracle masks has the potential of outperforming supervised Mask3D model on tail categories by 9.1% AP. The reviewer raised a concern about the fairness of our experiment, stating that Mask3D does not have access to oracle masks in test time unlike the open-vocabulary approaches presented in Tab. 3. We understand the concern, and we regret our unintendedly strong wording regarding the claim in L304-308. To address this, we conduct the suggested experiment in which we supply oracle masks to Mask3D. We perform Hungarian matching between the predicted masks and oracle masks discarding all class-losses, and only match based on the masks. For each oracle mask, we assign the class label of the matched mask from Mask3D. Our results are below:
>
> |Method|AP|head (AP)|common (AP)|tail (AP)|
> |---|---|---|---|---|
> |Mask3D (Hungarian M., oracle)|35.5|55.2|27.2|22.2|
> |OpenMask3D (oracle)|23.4|24.6|19.3|27.0|
>
> Even when we supply Mask3D with oracle masks, our approach surpasses the Mask3D performance on the tail categories by $+4.8$ AP. While these findings in fact confirm our initial claim, we will revise our text to make more careful conclusions.
>
> ### Performance OpenScene 2D Fusion vs. 2D/3D Ensemble
>
> The reviewer points out that OpenScene 2D performs better than the 2D/3D ensemble variant, which is not in line with the findings in OpenScene [46]. We also noticed this and show visualizations in Fig. 1.c of the rebuttal PDF: since the ensemble is a point-wise operator it seems to add inconsistent noise which might be the reason for the reduced performance.
>
> ### Limitations
>
> We agree with the reviewer that our paper would benefit from further discussing limitations, and we will extend our section (L320-323) to include a discussion on class-agnostic mask quality as also discussed in (L294-308). Although our findings above demonstrate that the 3D annotation bottleneck might not be as severe as initially thought, we understand that it still plays an important role and will be added to the limitations.

---

> > ### Comment · Reviewer_7yua · 2023-08-10
> >
> > Thank you so much for all your hard work and a thorough response!
> >
> > Generalization Experiments: I really appreciate these experiments and they resolve my concerns. For the scannet200 experiment, I am not sure though that this claim is well-supported: " Particularly, we see that OpenMask3D, compared to other open-vocabulary counterparts - seem to better preserve information about uncommon objects." since I think both OpenMask3D and OpenScene (2D fusion) see a similar drop i.e. around 3.3-3.6 in terms of mAP50 (All).
> >
> > Additionally, it might be useful to also add an additional row to Table-1 of rebuttal with a version where you DO have access to mask labels in Scannet 200 during training (the results you report in Table-1 of main paper). I think that would more easily illustrate the performance loss without access to ground truth mask labels.
> >
> > Thank you for fixing the oracle mask experiment!
> >
> > Overall, I would like to thank the authors for their effort; all my concerns are addressed. after seeing the rebuttal, I am leaning towards accept. I will update my ratings after the discussion with other reviewers.

---

> > > ### Author Response · Authors · 2023-08-14
> > >
> > > Thank you very much for your feedback, we are really happy to hear that we were able to address your concerns!
> > >
> > > Generalization Experiments: We are glad these experiments resolved the concerns. Regarding the claim "OpenMask3D, compared to other open-vocabulary counterparts - seem to better preserve information about uncommon objects.", we would like to clarify that what we originally meant was that OpenMask3D seems to perform better compared to other open-vocabulary counterparts in *this evaluation setup* as it achieved generally higher scores. However, we also agree that both methods see a similar drop in terms of AP50, and we will rephrase this statement to limit its scope to what we observe in this particular table.
> > >
> > > Furthermore, we agree that it would be helpful to extend Table 1 of the rebuttal with a row describing the results we reported in Table 1 of the main paper, to better illustrate the comparison between different experimental setups.
> > >
> > > Once again, thank you for your quick response and additional feedback! In the remaining discussion period, we would be very glad to address any additional questions that may arise, or any clarifications needed.

---

### Official Review · Reviewer_cqVz · 2023-07-04

**Soundness:** 3 good
**Presentation:** 4 excellent
**Contribution:** 3 good
**Rating:** 5
**Confidence:** 5

**Summary:**

This paper presents a method for 3D open-vocabulary instance segmentation. It proposes to use a class-agnostic Mask3D to get some instance mask proposals, project the instance points to 2D views to get some 2D segments, and extract some instance features based on the 2D segments by CLIP image encoder. Then, text descriptions can be used as queries to find the best instance proposals in an open-vocabulary way by comparing the similarities with the instance mask features.

**Strengths:**

1. The paper is well organized with good figures and easy to understand.
2. The paper is the first method for open-vocabulary instance segmentation in 3D scenes.
3. The experiments and analysis in the paper show the great zero-shot ability of the method.

**Weaknesses:**

1. The method relies on RGB-D images, and thus cannot generalize to 3D scenes (e.g., 3D scenes collected by LiDAR) without 2D images.
2. The experiments are conducted only on the ScanNet dataset. Experiments on more datasets like the Matterport3D used in OpenScene are expected to prove the generalizability of the method.
3. The inference speed of the whole framework seems slow as the method uses some heavy models (e.g., SAM and CLIP) for multiple views.  SAM model is even used multiple iterations for each mask. A comparison of the inference time between this method and OpenScene is expected.

**Questions:**

Please refer to the weaknesses. More questions are listed below:
1. More details of training a class-agnostic Mask3D are expected, for example, the scores used in the Hungarian matching process and the losses used in training.
2. The paper will be better if there is a discussion of the domain gap between the cropped image and the images used to train CLIP. Also, considering the domain gap, a detailed ablation of the hyperparameters in multi-scale cropping is expected.
3. The equations in L212 seem not accurate.



**Limitations:**

Some of the limitations are discussed in the last paragraph of the paper.

---

> ### Author Rebuttal · Authors · 2023-08-10
>
> We thank the reviewer for the feedback. We are happy that the reviewer found our paper well-organized and easy to understand, and appreciated the zero-shot ability of our open-vocabulary 3D instance segmentation approach. Below, we hope to answer the questions raised by the reviewer.
>
> ### W1/W2: Generalization to other datasets & 3D scenes without 2D images
>
> To assess the generalization ability of our method to other datasets, we share additional results on the Replica dataset. For the OpenScene baselines, we use the pre-computed features from the OpenScene repository. We provide qualitative examples of open-vocabulary queries from our method in Fig. 1.d of the rebuttal PDF. Our approach outperforms other open-vocabulary counterparts on the Replica dataset as shown below:
>
> |Method|AP|AP$_{50}$|AP$_{25}$|
> |---|---|---|---|
> |OpenScene (2D Fusion)|10.9|15.6|17.3|
> |OpenScene (3D Distill) |8.2|10.5|12.6|
> |OpenScene (2D/3D Ens.)|8.2|10.4|13.3|
> | OpenMask3D |**13.1**|**18.4**|**24.2**|
>
> Furthermore, the reviewer highlights that our approach requires images as input. This is correct, and it is due to the fact that OpenMask3D, like OpenScene, depends on visual-language models that operate on images in combination with text. In this work, we prioritized the ability to recognize uncommon/long-tail objects over generalization across different modalities. Using vision-language models on images provides an excellent opportunity to preserve this generalization capability. Nevertheless, we would like to state that when only a 3D scan of a scene is available, it could still be possible to render images from the 3D scan. We tried this on Replica dataset, and rendered RGB-D images from the scene point clouds, which is illustrated in Fig. 1.b of the rebuttal PDF. With this approach, we obtained the scores shown in the table below, marked "rendered RGB-D".
>
> |Method|AP|AP$_{50}$|AP$_{25}$|
> |---|---|---|---|
> |OpenMask3D (rendered RGB-D) |11.6|14.9|18.4|
> | OpenMask3D |**13.1**|**18.4**|**24.2**|
>
> Overall the performance does decrease when using images rendered from the point cloud, but only by -1.5 AP. Yet, we agree that when no color images are available or when the scan is sparse (LiDAR) it might not be possible to easily render color images.
>
>
> ### W3: Runtime
>
> In the table below, we provide the runtime of our approach on ScanNet for generating the queryable 3D scene representation, and also for performing a query on that representation. For comparison, we additionally provide the runtimes of the original OpenScene [46]. All runtimes are measured on the same hardware.
> | Method | Generating 3D Scene Representation [s] | Querying 3D Scene Representation [ms] | Semantics? | Instances? |
> | --- | --- | --- | --- | --- |
> | OpenScene (2D Fusion) |  440 | 168.8 | $\checkmark$| ✘  |
> | OpenScene (2D3D Ensemble) |  524 | 168.8 | $\checkmark$| ✘  |
> | OpenMask3D |  556 | **0.923** |  $\checkmark$| $\checkmark$ |
> | OpenMask3D (fast) |  **350** | **0.923** |  $\checkmark$| $\checkmark$ |
>
> We also report the runtime of a faster variation of our model (called ‘fast’) which runs CLIP only once per segment on a single crop and uses a smaller SAM backbone. The exact hyper-parameters are shown in the table below. This faster version performs almost as good as the original model (small performance drop of -1.2 AP).
>
> | Model | top k | $\mathsf{L}$ | SAM backbone |
> | --- | --- | --- | --- |
> | OpenMask3D |  5 | 3 |  ViT-H|
> | OpenMask3D (fast) |  1 | 1 |  ViT-B|
>
> |Method|AP|AP$_{50}$|AP$_{25}$|
> |---|---|---|---|
> |OpenMask3D (fast) |11.9|17.1|23.3|
> | OpenMask3D |**13.1**|**18.4**|**24.2**|
>
> ### Q1: Class-Agnostic Mask3D training details
> Class-Agnostic Mask3D closely follows Mask3D but ignores the semantics - main difference is how we discard class label and class confidence-based filtering stage. In the supplementary material Sec. 1.1 we provide a detailed explanation. We will extend the description in the main text to include further implementation details.
>
> ### Q2: Discussion on the domain gap & ablation on multi-scale cropping parameters
> We agree with the reviewer, and provide an ablation study regarding the hyperparameters used during the multi-scale cropping phase on the Replica dataset. In the table below, we ablate results both based on number of levels, and ratio of expansion between levels:
>
> | Levels | Ratio of Expansion|AP | AP50| AP25|
> | --- | --- | --- |--- |--- |
> | 1 | 0.1 |11.3 |16.0|20.2|
> | 3|  0.1 | **13.1**|**18.4**|**24.2**|
> | 5| 0.1 |12.8|17.6|22-6|
> | 3|  0.05| 12.9 |18.1 |23.5 |
> | 3|  0.1 |**13.1**|**18.4**|**24.2**|
> | 3|  0.2| 12.8 | 17.7 | 22.9 |
>
> Regarding the domain gap in w.r.t. CLIP, we also would like to highlight our "rendered RGB-D" experiment on Replica discussed earlier, which we believe highlights the robustness of our CLIP-based mask-feature module to image-quality and realism.
>
>
> ### Q3: Equations in L212
> The reviewer noticed a problem with the equations, we updated them as follows:
>
> - $x_1^l = \max(0, x_1^1-(x_2^1-x_1^1)\cdot k_{exp} \cdot l)$
> - $y_1^l = \max(0, y_1^1-(y_2^1-y_1^1)\cdot k_{exp}\cdot l)$
> - $x_2^l = \min(x_2^1+(x_2^1-x_1^1)\cdot k_{exp}\cdot l, W-1)$
> - $y_2^l = \min(y_2^1+(y_2^1-y_1^1)\cdot k_{exp}\cdot l, H-1)$

---

> > ### Author Response · Authors · 2023-08-21
> >
> > Dear reviewer cqVz,
> >
> > As the discussion phase ends today we will not be able to further clarify potential additional concerns. We would be very grateful if you could respond to our rebuttal and offer us an opportunity to further engage with your concerns and address any additional questions you might have!
> >
> > Thank you for your time and feedback!
> >
> > Best,
> >
> > Authors

---

### Official Review · Reviewer_zuiw · 2023-07-07

**Soundness:** 3 good
**Presentation:** 3 good
**Contribution:** 3 good
**Rating:** 4
**Confidence:** 3

**Summary:**

This paper addresses the problem of text-based 3D instance segmentation. To tackle this problem, the authors propose OpenMask3D, a zero-shot approach for 3D instance segmentation that utilizes class-agnostic 3D instance masks and multi-view fusion of CLIP-based image embeddings to aggregate per-mask features. The model's performance is evaluated through experiments and ablation studies on the ScanNet200 dataset, where it outperforms OpenScene in some metrics.


**Strengths:**


**Clarity and quality**: The paper is well written and explains all the components with clarity and in detail. The figures are excellent. The results are very neatly presented and overall the paper is engaging to read.

**Empirical Evaluation**: The paper includes extensive empirical evaluations on the ScanNet200 dataset. The paper not only demonstrates the performance of OpenMask3D but also presents ablation studies on the number of frames used, the use of 2d masks and multi-scale crops to understand the contribution of these components.

**Relevance**: The work addresses a highly relevant challenge in computer vision and autonomous systems. As these technologies become increasingly prevalent, the ability to understand and interact with a diverse range of objects becomes critically important, underscoring the relevance of this research.

**Weaknesses:**


**Novelty of the task**: The paper makes strong claims about being the first to introduce "open-vocabulary 3D instance segmentation". However, given OpenScene and others I don't think it can be claimed that the problem of text-based 3D instance segmentation is novel. OpenScene might not have used the word "instance" but it does show results for "Open-vocabulary 3D object search" which is similar in meaning.

**Significance of the Contribution**:  It is true that most existing segmentation approaches have relied on a closed-set of objects in 3D annotated datasets and thus a system that can perform open-vocabulary segmentation would be quite a significant contribution to the field of 3D scene understanding. However, the technical contribution of the proposed approach is somewhat limited considering its similarity to OpenScene. The main differences seem to be the frame selection and feature aggregation strategy, but if this is the case I would expect these components to be more prominently handled in the paper (rather than claiming a whole novel task).

**Questions:**


- Please clearly outline the differences compared to OpenScene
- The examples given in the paper are mainly for common objects "armchair", "seat", "pool", "sofa". Can you give an example where the method identifies objects that are truly from an "open" vocabulary (i.e. not common)?

---

> ### Author Rebuttal · Authors · 2023-08-09
>
> We thank the reviewer for providing valuable feedback, and we are really glad that the reviewer found our paper enjoyable to read! We address the questions and concerns in the following:
>
> ### Q1: Differences between OpenMask3D (instance segmentation) and OpenScene (semantic segmentation)
>
> The reviewer asks about the differences between OpenScene and this work, in particular the reviewer suggests that OpenScene already solves the task of "open-vocabulary 3D instance segmentation".
> We disagree: OpenScene mainly addresses *semantic* segmentation (i.e., it predicts per-point features), while OpenMask3D addresses *instance* segmentation (i.e., it predicts a set of object masks and associated features). Both tasks are fundamental computer vision problems, that are well defined and conceptually different. "Open-vocabulary 3D object search" in OpenScene primarily identifies a single point in the scene that matches the query the best. "Image-based 3D object detection" in OpenScene identifies the set of points that have close similarity to the given image-based query (e.g. a chair image), however it cannot differentiate between multiple instances of the same object class (e.g., it cannot predict a segmentation mask for Chair 1 and a different mask for Chair 2).
>
> In the light of this, we would like to clearly outline the main conceptual differences between OpenMask3D and OpenScene. OpenMask3D is *mask-based*, while OpenScene is *point-based*. Specifically, OpenScene outputs a per-point feature representation (num_points, feature_dim). OpenMask3D outputs a set of binary instance masks (num_masks, num_points), and corresponding per-mask features (num_masks, feature_dim). We would like to draw the reviewer’s attention to Fig. 1.a and Fig. 1.c in the rebuttal PDF, and Fig. 7 in the supplementary material, which we believe are helpful for visualizing our following statements.
>
> Both OpenMask3D and OpenScene output task-agnostic open-vocabulary features, but our method is tailored towards identifying object *instances*. While OpenScene returns heatmaps over the scene points describing similarity to the query, it cannot differentiate between two object instances that belong to the same query. In practice, it can either retrieve a set of points from the point cloud by using a given similarity score threshold (Sec. 5 of OpenScene paper, “Image-based 3D object detection”) or can identify a single point that best matches the query (Sec. 5 of OpenScene paper, “Open-vocabulary 3D object search”). In contrast, OpenMask3D is able to segment the *object instances* that suit the given open-vocabulary query, automatically separating the multiple instances from each other.
>
> Suppose now we want to find top-k objects which match an open-vocabulary query. Although, as the reviewer correctly underlined, OpenScene is able to perform this task for $k=1$ (Sec, 5 of OpenScene paper, "Open-vocabulary 3D object search"), it cannot retrieve multiple object matches when $k>1$, because it cannot distinguish whether a point belongs to a specific object or to another one.
>
> Suppose finally we want to find how many objects are present in a room such that the similarity with a given query is higher than a given threshold. In this case OpenScene has no direct way to provide such an answer as it can only return a per-point features/labels.
>
> In brief, OpenScene and many recent 3D open-vocabulary approaches need consequent steps to identify and separate object instances, which poses many practical limitations. As the reviewer also highlighted, “the ability to understand and interact with a diverse range of objects becomes critically important” - particularly when coupled with the necessity to densely segment separate objects. The novelty in our work lies in its ability to directly identify object instances in an open-vocabulary setting, and its design that is tailored to efficiently focus on *instances*.
>
> **OpenScene uses OpenSeg, OpenMask3D uses CLIP**
>
> This is another fundamental difference between both methods: to obtain pixel-aligned CLIP features, OpenScene relies on OpenSeg which is a fine-tuned version based on CLIP. However, during fine-tuning, it becomes less "general" than the original CLIP (used by OpenMask3D). This effect is also shown by the results in Table 1 of the main paper, particularly in long-tail categories.
>
> ### Q2: Examples from an “open” vocabulary
>
> The reviewer asks whether we could share examples where the method identifies objects that are truly from an “open” vocabulary. Our method is indeed capable of segmenting uncommon object instances, and we would like to draw attention to several figures from our submission. In Fig. 1 (main paper), we share examples using the queries “footrest” and “angel”. In Fig. 4 of the main paper, we share additional “uncommon” object queries, such as “pepsi”. We also would like to highlight sup. mat. Fig. 8, in which *all* examples are uncommon categories. This figure highlights examples such as “Cetaphil” (a soap brand), “Roomba” (a robot vacuum cleaner brand) and “dollhouse”. We apologize if we have not referenced the figures sufficiently within the text to highlight particularly novel object queries.

---

> > ### Author Response · Authors · 2023-08-21
> >
> > Dear reviewer zuiw,
> >
> > As the discussion phase ends today we will not be able to further clarify potential additional concerns. We would be very grateful if you could respond to our rebuttal and offer us an opportunity to further engage with your concerns and address any additional questions you might have!
> >
> > Thank you for your time and feedback!
> >
> > Best,
> >
> > Authors

---

### Official Review · Reviewer_X8Ds · 2023-07-21

**Soundness:** 2 fair
**Presentation:** 3 good
**Contribution:** 2 fair
**Rating:** 4
**Confidence:** 4

**Summary:**

The paper proposes to solve open-vocabulary 3D instance segmentation. It uses a class-agnostic 3D instance segmentation model to obtain instance masks, then gather multi-scale image features from multiple frames by CLIP and SAM to do the open-vocabulary classification task.

**Strengths:**

1. The paper proposes to use instance-level features for 3D open-vocabulary instance segmentation, which is not attempted by previous works
2. The proposed module exhibits reasonable improvements, as shown in Tables 2&3.

**Weaknesses:**

1. The design of the framework may be complicated for real-world usages. It uses SAM to do segmentation for multiple frames, and then use multi-scale images for CLIP to inference. Each component like SAM and CLIP is a large foundation model and takes a while to inference, not mention that they are used multiple times.
2. The idea is not so novel. Although such idea of using instance mask features is not attempted in 3D instance segmentation, it has been widely adopted in 2D open-vocabulary segmentation tasks, like OpenSeg[1], ODISE[2], ZegFormer[3].
3. The experiments are not thorough. For example, the details of using SAM and RANSAC are not studied.


[1] Scaling Open-Vocabulary Image Segmentation with Image-Level Labels, ECCV2022
[2] ODISE: Open-Vocabulary Panoptic Segmentation with Text-to-Image Diffusion Models, CVPR2023.
[3] Decoupling Zero-Shot Semantic Segmentation.


**Questions:**

1. What is the latency and inference cost of the model during inference?
2. How $k_{rounds}$ is chosen? Is there any ablation study about that? Furthermore, does it mean that SAM needs to inference $k_{rounds}\times number of frames$, and CLIP need inference $k_{rounds}\times number of frames \times number of scales$
3. How the confidence score $s_r$ is obtained?

**Limitations:**

See weakness

---

> ### Author Rebuttal · Authors · 2023-08-09
>
> We thank the reviewer for the valuable feedback.
>
> ### Q1: What is the latency and inference cost of the model during inference?
>
> In the table below, we provide the runtime of our approach on ScanNet for generating the queryable 3D scene representation, and also for performing a query on that representation. For comparison, we additionally provide the runtimes of the original OpenScene [46]. All runtimes are measured on the same hardware.
> | Method | Generating 3D Scene Representation [s] | Querying 3D Scene Representation [ms] | Semantics? | Instances? |
> | --- | --- | --- | --- | --- |
> | OpenScene (2D Fusion) |  440 | 168.8 | $\checkmark$| ✘  |
> | OpenScene (2D/3D Ensemble) |  524 | 168.8 | $\checkmark$| ✘  |
> | OpenMask3D |  556 | **0.923** |  $\checkmark$| $\checkmark$ |
> | OpenMask3D (fast) |  **350** | **0.923** |  $\checkmark$| $\checkmark$ |
>
> We also report the runtime of a faster variation of our model (called "fast") which runs CLIP only once per segment on a single crop and uses a smaller SAM backbone. The exact hyper-parameters are shown in the table below. This faster version performs almost as good as the original model (small performance drop of -1.3 AP$_{50}$, full table is available as Tab. 2 of the rebuttal PDF).
>
> | Model | top-k | $\mathsf{L}$ | SAM backbone |
> | --- | --- | --- | --- |
> | OpenMask3D |  5 | 3 |  ViT-H|
> | OpenMask3D (fast) |  1 | 1 |  ViT-B|
>
> ### Q2.a: Value of $k\_{rounds}$
>
> The reviewer asks about how $k\_{rounds}$ (number of SAM runs in RANSAC) is selected and if there is an analysis study.
> We tried increasing values for $k\_{rounds}$ but saw only a marginal increase in performance, e.g, from 16.6 AP50 (k=1) to 16.8 AP50 (k=10).
> Interestingly, increasing the number of SAM runs has little impact on the runtime performance.
> Indeed, the table below shows (for SAM with different backbones), that the high computational cost of SAM comes when the image is set (which is done only once) and only little overhead is added when re-running the prediction based on a new set of ground truth points. Thus, performing multiple SAM predictions on the same image has no large impact on the runtime, but makes our model more robust.
>
> | Function | Backbone | time [s] |
> | --- | --- | --- |
> | SAM.set_image() |  ViT-H | 0.497 |
> | SAM.predict() |   ViT-H  | 0.006 |
> | SAM.set_image() |   ViT-B | 0.109 |
> | SAM.predict() |   ViT-B | 0.005 |
>
> ### Q2.b: How often is SAM and CLIP called?
>
> Since we only use the top k_view views for each mask, not all frames are utilized.
> Therefore, the number of iterations for SAM is $\mathsf{k_{rounds}\cdot \mathbf{M \cdot k_{view}}}$ and for CLIP, it is $\mathsf{\mathbf{M \cdot k_{view}}\cdot L}$, where $\mathsf{M}$ is the number of 3D masks predicted by the class agnostic mask predictor and $\mathsf{L}$ is the number of levels used for the multi-level crop. Overall this approach is comparable in runtime to OpenScene which needs to run OpenSeg over the full frame (which is slow), whereas here we run the much faster CLIP on small crops.
>
> ### Q3: How is the confidence score $s_r$ obtained?
>
> The confidence score $s_r$ is an output of the SAM model. For each 2D mask, SAM also predicts a confidence score $s_r$ (see main paper L201, L204-207). We will describe this more clearly in the paper. In the supplementary material (Sec, 1.2.), we provide further explanations on how we utilize SAM (L52-L92), and visualize 2D mask proposals along with confidence scores returned by SAM in Fig. 3 to Fig. 6.

---

> > ### Author Response · Authors · 2023-08-21
> >
> > Dear reviewer X8Ds,
> >
> > As the discussion phase ends today we will not be able to further clarify potential additional concerns. We would be very grateful if you could respond to our rebuttal and offer us an opportunity to further engage with your concerns and address any additional questions you might have!
> >
> > Thank you for your time and feedback!
> >
> > Best,
> >
> > Authors

---

### Author Rebuttal · Authors · 2023-08-09

We thank all the reviewers for their valuable feedback, we appreciate their detailed suggestions. We reply to each reviewer’s questions and concerns in the individual responses, and we have added tables and figures in the attached rebuttal PDF, which we reference and explain in the responses.

Here, we also would like to provide an overview of the material in the attached document:

**Table 1.** We provide 3D instance segmentation results using masks from mask module trained on ScanNet, evaluated on the ScanNet200 dataset. We identify classes (such as chair, folded chair, table, dining table ...) that are semantically close to the original ScanNet classes, and group them as “Base”. Remaining classes are grouped as “Novel”. We also report results on the full set of labels.

**Table 2** - We provide additional experiments on the **Replica** dataset. We further analyze different training setups quantitatively.

**Table 3** - We provide an ablation study on the hyperparameters related to the multi-scale cropping stage of our approach.

**Table 4 & Table 5** -  We provide an overview of the memory requirements of foundation models used in OpenMask3D and OpenScene, and the time requirements for atomic operations.

**Figure 1.a & Figure 1.c** - Qualitative comparison of per-mask (OpenMask3D) and per-point (OpenScene) features.

**Figure 1.b** - Illustration of our additional experiment on the Replica dataset, in which we render RGB-D images from the point clouds, and use these images as an input to our pipeline.

**Figure 1.d** - Qualitative results from open-vocabulary queries on the Replica dataset, using our OpenMask3D approach.

---

### Decision · Program_Chairs · 2023-09-21

**Decision:**

Accept (poster)

**Comment:**

This paper initially received mixed reviews.  An initially negative reviewer upgraded their opinion to positive in light of the clarifications and details in the authors' rebuttal responses.  Two other reviewers who were positive indicated that the discussion period alleviated their concerns and their post-rebuttal opinion remains positive with respect to acceptance.  One negative remaining reviewer expresses a concern regarding novelty that the AC does not find to be substantiated with sufficient specificity, while another negative reviewer has not fully engaged in the discussion period or expressed an opinion after the author rebuttal.  Therefore, the AC does not find a strong basis to reject the paper.  The AC recommends that the authors incorporate the clarifications and additional results provided during the discussion period when preparing the camera ready version of the paper.